# A novel mRNA-based multiepitope vaccine candidate against *Cryptosporidium hominis* and *Cryptosporidium parvum* employing reverse-vaccinology and immunoinformatics approaches

Ahmad Abdullah Mahdeen[1☯], Imam Hossain[2☯*], Md. Habib Ullah Masum[3], T. M. Fazla Rabbi[2], Sajedul Islam[2]

1 Department of Microbiology, Notre Dame University Bangladesh (NDUB), Arambagh, Motijheel, Dhaka, Bangladesh, 2 Department of Microbiology, Noakhali Science and Technology University, Noakhali, Bangladesh, 3 Department of Genomics and Bioinformatics, Faculty of Biotechnology and Genetic Engineering, Chattogram Veterinary and Animal Sciences University (CVASU), Khulshi, Chattogram, Bangladesh

☯ These authors contributed equally to this work

* ihossain.mbg@nstu.edu.bd

## Abstract

### Introduction

The parasite *Cryptosporidium* spp. causes cryptosporidiosis, a diarrheal disease in humans and animals. This study describes the development of mRNA vaccine targeting antigens from *C. hominis* and *C. parvum*, important gut pathogens. This vaccine was designed with reverse vaccinology and immunoinformatics as no FDA-approved vaccine exists for cryptosporidiosis.

### Materials and methods

Initially, a thorough literature review was conducted to identify five pathogenic proteins (aminopeptidase, heat shock protein, P23, serine protease, and sporozoite glycoproteins) associated with these two parasites. Next, a multiple sequence alignment was conducted, and the conserved sequences were used to design a novel multiepitope mRNA vaccine against these two parasites, combining the best CD8+, CD4+, and continuous B-cell epitopes. Additionally, structural prediction, docking, dynamics, and immune simulation, as well as cloning, were conducted.

### Result

The vaccine demonstrated standard biophysical properties, indicating that the protein is soluble and stable. Both two-dimensional (substantial alpha helix, beta sheet, and coil structures) and three-dimensional structures (Ramachandran score of 83.1% and a Z score of −7.39) of the vaccine were standard. The docking energy for TLR-2

**Data availability statement:** All relevant data are within the paper and its Supporting Information file.

**Funding:** The author(s) received no specific funding for this work.

**Competing interests:** The authors have declared that no competing interests exist.

(−1151.9) and TLR-4 (−1028.3) exhibited significant interactions. Furthermore, MM-GBSA and dynamics simulation both verified their stability, compactness, and flexibility. Next, codon optimization for *Escherichia coli* expression yielded promising results, with the vaccine demonstrating substantial expression, as evidenced by a GC content of 46.97% and a CAI of 0.988. Afterwards, immune simulation demonstrated robust immune response amplification upon repeated exposures. In addition, the vaccine exhibited stability in its mRNA structures.

## Conclusion

This study developed an *in-silico* multiepitope novel mRNA vaccine candidate for *C. hominis* and *C. parvum* with excellent structural stability, antigenicity, receptor-binding affinity, and expected immune responses. These findings offer a novel approach due to numerous species target but with significant drawbacks like no validation beyond simulation, uncertainty of long-term immunity, protein quality, stability and safety, requiring experimental validation.

## 1. Introduction

Cryptosporidiosis is an ubiquitous, parasitic, zoonotic and opportunistic infection caused by *Cryptosporidium* spp that can be transmitted between animals and humans [1]. There are a lot of species of *Cryptosporidium* but *Cryptosporidium hominis* and *Cryptosporidium parvum* cause infection in humans frequently [2]. Humans are the main natural host of *C. hominis* while *C. parvum* infects both bovines and humans [3]. This parasite causes the infection by infecting the microvillous region of epithelial cells lining the digestive and respiratory organs of vertebrates significantly contributing to 20% cause of diarrhea in humans and animals and also poses a significant risk to individuals with weakened immune systems, particularly those with AIDS [4,5]. Moreover, up to 24% of individuals globally with both AIDS and diarrhea are affected by this parasite [6]. The common symptoms are diarrhea, stomach cramps, nausea, vomiting, fever, weight loss, dehydration etc and the severe symptoms are chronic cryptosporidiosis, life-threatening problems with digesting foods and absorbing nutrients [7].

Cryptosporidiosis was initially identified in humans during the late 1970s, predominantly impacting immunocompromised individuals, with a significant rise in cases noted at the onset of the AIDS epidemic in the early 1980s [8]. Since 2005, reported cases in the United States have increased significantly, partly attributable to the emergence of the *C. hominis* subtype IaA28R4 [8]. Surveillance data reveal significant fluctuations in incidence, with marked rises in outbreaks and related hospitalizations across several states, especially in the Great Lakes region. Beyond human health, cryptosporidiosis continues to be a significant issue in veterinary medicine, with bovine infections representing a primary source of calf enteritis globally, thereby exerting substantial economic pressures on livestock production systems [9,10]. Given the need, this study focuses on designing a potential vaccine candidate to overcome the effects of cryptosporidiosis.

The comparison of the genomes of *C. hominis* and *C. parvum* revealed a high degree of similarity, with just a 3–5% difference in their genetic sequences in terms of major insertions, deletions, or rearrangements. Indeed, the gene complements of the two species are virtually indistinguishable, as the small number of *C. parvum* genes that are absent in *C. hominis* are located around known gaps in the sequence [11].

Cryptosporidium has a straightforward life cycle since it only needs one host to complete its asexual and sexual stages, and it infects cells lining the intestines [12]. It progresses through six distinct stages of development [13]. The life cycle of the parasite progresses through several distinct stages. First, excystation occurs as stomach acids or intestinal enzymes break the oocyst's tough outer shell, releasing infective sporozoites. These sporozoites then enter intestinal epithelial cells, initiating merogony, a process of asexual reproduction that increases the parasite population. Following merogony, some merozoites develop into male or female gametocytes, initiating gametogony. If the sexual cycle occurs, microgametocytes produce motile male gametes, which fertilize macrogametes, resulting in zygote formation. The fertilized zygote encysts and forms a thick wall, leading to the formation of an oocyst wall. These environmentally resistant oocysts can be shed in feces, allowing the parasite to resume its life cycle. Finally, sporogony occurs within the fertilized oocyst, completing the cycle [14]. Cryptosporidium infection results from ingesting a tiny number of oocysts. However, the infectious dose for primate can be as little as 10 oocysts [15]. Notably, the parasite attachment to host cells is a crucial first stage in the establishment of infection. It has been determined and demonstrated that two kinds of proteins—mucin-like glycoproteins and thrombospondin-related adhesive proteins—mediate adhesion [14]. By adhering to or invading the intestinal epithelial cells, or by generating toxins that increase intestinal secretion, infectious organisms can change intestinal ion transport. The infection might potentially change the ability of intestinal epithelial cells to transport substances and/or trigger a neurohormonal, endocrine, or immunological response that modifies the ratio of intestinal absorption to secretion [16]. However, there are several proteins of the parasites that aid in pathogenesis [17]. To cite an example, aminopeptidases are the proteins that break down host peptides into smaller units for nutrient acquisition and also can modulate the immune system of the host [17]. Next, heat shock proteins, especially heat shock proteins 70 and 90, are expressed under stressful conditions, such as oxidative stress or heat, and help the parasite maintain protein folding, prevent denaturation, and modulate the host's immune response [18,19]. Subsequently, the P23 is the protein for aiding the parasite to attach to the enterocyte of the host intestine, initiating infection [20]. The serine proteases play a role in modulating host immune response and extracellular matrix breakdown for host tissue penetration and replication [21,22]. The sporozoite glycoproteins are also critical for host cell attachment, immune modulation, and invasion [23].

There is no approved vaccine against cryptosporidiosis, and the role of antiparasitic medications in its treatment is unclear. The clinical and parasitological responses to treatment vary significantly with the patient's immune status, the duration of treatment, and the concomitant use of other medications [24]. However, Nitazoxanide demonstrated efficacy in treating cryptosporidiosis in individuals with a normal immune system and likely in those with a damaged immune system. Nevertheless, it is insufficient as a standalone treatment for the condition [25]. Notably, vaccinations can provide effective treatment for cryptosporidiosis. Vaccines are essential for maintaining global health because they stop the spread of many diseases and infections by stimulating the immune system [26]. From Edward Jenner (1796) to SARS-CoV-2 (2019), vaccination has saved the lives of millions of people [27]. Compared with other types of vaccines, including subunit, killed, live-attenuated, toxoid, viral vector-based, and DNA-based vaccines, mRNA-based vaccines offer several advantages, including stability, affordability, and lack of integration into DNA [28–31]. The mRNA vaccine and reverse vaccinology (RV) methods received substantial attention when the FDA granted emergency use authorization for the initial SARS-CoV-2 mRNA vaccines, BNT162b2 (Pfizer/BioNTech) and mRNA-1273 Spikevax (Moderna). In recognition of their groundbreaking work in mRNA technology, Professors Drew Weissman and Katalin Karikó were awarded the Nobel Prize in Physiology or Medicine in 2023 for their pivotal contributions to vaccine development [32–34].

Nowadays, vaccine is constructed using RV and immunoinformatics approaches [35]. The term "RV" was coined to describe the approach of using computer-based genetic information to initiate vaccine research, rather than working

directly with the pathogen in a laboratory [35]. The strategic development of vaccines targeting specific epitopes, using AI for both B- and T-cells, is a critical step in this process. This method provides a potent, immunogenic vaccine that confers protection against autoimmunity and other adverse outcomes across different ethnicities, pregnant women, and susceptible age groups [36].

At present, there is no efficacious vaccine available to prevent bovine cryptosporidiosis. However, a recombinant *C. parvum* gp40 protein was evaluated as a vaccine in pregnant heifers during late gestation where vaccinated cows exhibited significantly superior health, with reduced severity and duration of diarrhea, as well as enhanced weight gain compared to controls indicating, good efficacy of the vaccine [37].

Consequently, the aim of this study was to develop a novel mRNA vaccine candidate for the treatment of cryptosporidiosis using RV, as no mRNA vaccine is currently approved by the FDA. This current vaccine will probably elicit good immune responses against two potent species of cryptosporidiosis. The efficacy of mRNA vaccines and RV served as a substantial impetus for us, as both approaches facilitate the identification of ideal vaccine constituents and yield safe, stable and effective vaccines [33,34,38,39]. However, the research gaps of this study are of no validation beyond simulation, uncertainty of long-term immunity, protein quality, stability and safety, requiring experimental validation.

## 2. Materials and methods

### 2.1. Obtaining of sequence and alignment

The amino acid sequences of the aminopeptidase, heat shock protein, P23 protein, serine protease, and sporozoites glycoprotein of the *C. hominis* with accession numbers of XP_666607.1, V9QET8, Q5CHW5, PPS97477.1, and Q4VY80, respectively, were obtained employing the National Center for Biotechnology Information (NCBI) and saved in FASTA [40]. The same procedure was followed for the same proteins of *C. parvum* with accession numbers of A0A7S7LGW5, B5STM2, F8UW66, A0A7G2HIR5, and Q4VY88, respectively. Next, the consensus sequences were generated from all the aminopeptidase, heat shock protein, P23 protein, serine protease, and sporozoite glycoprotein sequences of both parasites using the BioEdit 7.2 program [41]. Subsequently, all targeted protein sequences were saved as individual FASTA files for the next steps.

### 2.2. Prediction of CD8+ (MHC-I), CD4+ (MHC-II) and continuous B-cell epitopes

The Immune Epitope Database (IEDB) (http://tools.immuneepitope.org/mhci/) was used to prediction the CD8+ and CD4+binding epitopes for the conserved sequences of aminopeptidase, heat shock protein, P23 protein, serine protease, and sporozoite glycoprotein in both parasites [42–54]. This server is a detailed database offering details regarding antibodies, B-cell epitopes, T-cell epitopes, MHC molecules, and MHC binding ligands in humans and numerous animal species, including primates and mice [42–55]. Subsequently, the 9-mer, 15-mer and <1.00 percentile ranked epitopes (threshold) were selected for subsequent analysis, respectively. Moreover, the continuous B-cell epitopes of the targeted conserved sequences from the same proteins were predicted using the ABCpred (https://webs.iiitd.edu.in/raghava/abcpred/) server [56]. This server employs neural network technology for predicting MHC-II binding epitopes. [57,58]. Afterwards, the antigenicity, allergenicity and toxicity were predicted employing VaxiJen 2.0 (http://www.ddg-pharmfac.net/vaxijen/VaxiJen/VaxiJen.html) (threshold 0.4), AllerTOP v.2.0 (https://www.ddg-pharmfac.net/AllerTOP/index.html) (threshold <0.00) and ToxinPred (https://webs.iiitd.edu.in/raghava/toxinpred/algo.php) servers (threshold <0.00), respectively [59–61]. VaxiJen 2.0, AllerTOP v.2.0, and ToxinPred have exhibited reliable accuracy in predicting bacterial and viral antigens, with performance varying depending on the particular pathogen. Nonetheless, the accuracy of these tools depends on the quality and diversity of the dataset employed during training. Predictions may be less dependable for proteins that do not resemble known antigens or possess atypical folds [62].

## 2.3. Analysis of the population coverage of the selected epitope

The development of a multiepitope vaccine is profoundly shaped by population coverage analysis, as the high polymorphism of HLA alleles restricts the proportion of individuals capable of responding to a specific antigen; consequently, analyzing the population coverage for T-cell epitopes reveals their potential to stimulate T-cell responses in a substantial number of individuals across diverse geographical regions [63–66]. However, the population coverage of the selected epitopes was determined using the Population Coverage Analysis tool by IEDB (http://tools.iedb.org/population) with the standard threshold of >97% for MHC-I and >99% for MHC-II epitopes and a standard 27 reference set of human HLA alleles [67]. The coverage of populations from all regions of the globe was also determined [68,69]. Based on HLA (Human Leukocyte Antigen) frequencies, binding predictions (MHC-I & II), and T-cell response data, this tool predicts the proportion of specified worldwide populations projected to respond to a set of epitopes. Using allele frequencies from external databases and binomial distribution and Hardy-Weinberg to determine genotype frequencies, it identifies binders for different HLA types and sums population coverage, outputting fractions recognizing zero, one, or multiple epitopes and vaccine design metrics like PC90 (epitopes recognized by 90% of the population) [68,69].

## 2.4. Construction of vaccine candidate

Vaccine construction was achieved by targeting highly prioritized epitopes from conserved sequences of aminopeptidase, heat shock protein, P23 protein, serine protease, and sporozoite glycoprotein of both parasites. Linkers such as EAAAK were implemented for ligating the CD8+ epitopes with adjuvant 50S ribosomal protein L7/L12, AYY was used for ligating CD8+ epitopes, AK was used for ligating the CD4+ epitopes, and KFER was used for ligating continuous B-cell epitopes [70–73].

## 2.5. Validation of biophysical qualities

The Expasy ProtParam server (http://web.expasy.org/protparam/) provided an analysis of the vaccine's biophysical properties, including the number of atoms, molecular mass, net charge, instability, aliphatic index, isoelectric point, and grand average of hydropathicity (GRAVY) [74]. Next, the allergenicity was evaluated by AlgPred (https://webs.iiitd.edu.in/raghava/algpred/submission.html) [75], AllerTOP v. 2.0 (https://www.ddg-pharmfac.net/AllerTOP/) [61] and AllergenFP v.1.0 (http://ddg-pharmfac.net/AllergenFP/) servers [76,77]. The antigenicity was determined by VaxiJen 2.0 (http://www.ddg-pharmfac.net/vaxijen/VaxiJen/VaxiJen.html) servers [60] and ANTIGENpro (http://sc.ratch.proteomics.ics.uci.edu) servers [78].

## 2.6. Modelling of two-dimensional structure

The PSIPRED (http://bioinf.cs.ucl.ac.uk/psipred/), GOR4 (https://npsa-prabi.ibcp.fr/cgi-bin/npsa_automat.pl?page=/NPSA/npsa_gor4.html) and the SOPMA (https://npsa-prabi.ibcp.fr/cgi-bin/npsa_automat.pl?page=/NPSA/npsa_sopma.html) were utilized to generate the two-dimensional structure [79–82]. The PSIPRED server utilizes two feed-forward neural networks to evaluate the output from PSI–BLAST (Protein Specific Iterated Basic Local Alignment Search Tool) for the prediction of two-dimensional structure [83]. Moreover, the GOR4 utilizes the theory of information and Bayesian statistical methods in order to predict the secondary structure of a protein. [84]. Conversely, the SOPMA server is capable of predicting approximately 69.5% of amino acids within a protein sequence concerning a three-state representation (alpha-helix, beta-sheet, and coil) of the two-dimensionalstructure [81]. The accuracy of PSIPRED, GOR4, and SOPMA in predicting two-dimensionalstructures generally falls between 75% and 85%. PSIPRED and GOR4 usually exhibit superior accuracy; however, all three tools may encounter difficulties with intrinsically disordered regions, flexible structures, or sequences exhibiting low homology to known proteins, which may result in less reliable predictions in such instances [62].

## 2.7. Modelling, refinement and validation of three-dimensional structure

The I-TASSER server (https://zhanggroup.org/I-TASSER/) was utilized for modelling the three-dimensional structure of the vaccine [54,85]. This server conducts multiple threading alignments and iterative template fragment assembly simulations to anticipate the three-dimensional structure of a protein [54,85]. Subsequently, the refinement was conducted on GalaxyWEB (https://galaxy.seoklab.org/cgi-bin/submit.cgi?type=REFINE) server [86]. Additionally, the validation of structure was conducted on SAVES v6.0 (https://saves.mbi.ucla.edu/) for the Ramachandran and ERRAT score analysis and ProSA server (https://prosa.services.came.sbg.ac.at/prosa.php) for the Z score analysis in terms of protein quality [87–92]. The reliability of SAVES v6.0 and ProSA in assessing protein models is generally sound; however, these tools may fail to identify nuanced errors, especially in flexible or atypical protein regions. Their predictions should be interpreted alongside experimental data for enhanced accuracy [62].

## 2.8. Prediction of discontinuous B-cell epitopes

For the prediction of discontinuous B-cell epitopes, the Ellipro (http://tools.iedb.org/ellipro/) server was utilized [93,94]. This server predicted the epitopes through surface accessibility analysis and the integration of three algorithms [93]. Nonetheless, the precision of this tool is contingent upon the integrity of the input protein structure. If the structure is inaccurately predicted or lacks high-resolution data, the reliability of epitope prediction may diminish [62].

## 2.9. Docking interaction analysis

The ClusPro 2.0 server (cluspro.bu.edu/login.php) was employed for molecular docking analysis [95–98]. First, three-dimensional structures of human TLR-2 (PDB: 2Z7X) and TLR-4 (PDB: 3FXI) were selected from the Protein Data Bank (PDB) database (www.rcsb.org). Subsequently, the visualization and analysis of the complexes were conducted in BIOVIA Discovery Studio Visualizer (https://discover.3ds.com/discovery-studio-visualizer-download) and PDBsum (http://www.ebi.ac.uk/thornton-srv/databases/pdbsum/Generate.html) [99–101].

## 2.10. Dynamics simulation of the molecules

The dynamics simulation of the molecules elucidates the coordinated functional movements of macromolecules [102,103]. Therefore, the iMODS (http://imods.chaconlab.org/) server was applied to run a dynamics simulation of molecules for the vaccine-TLR-2 along with vaccine-TLR-4 complexes [103]. The server offers an extensive array of robust motion configurations, including affine-model arrows, vector fields, and modal animations. Moreover, this server calculates a range of properties, including mobility (B-factor), deformability, eigenvalues, covariance maps, and linkage matrices [103]. The B-factor elucidates the extent to which atoms deviate from their structural equilibrium [100]. Moreover, the deformability graph serves as a graphical representation of flexibility of protein, particularly for coil or domain linkers [104]. Additionally, the eigenvalue serves as a criterion for the stability of the complex, with elevated values signifying a greater degree of stability [103,105].

## 2.11. Post-simulation analysis (MM-GBSA)

Molecular mechanics employing the generalized Born and surface area solvation (MM-GBSA) approaches was employed to compute the negative free energy associated with the interactions of docked complexes [106]. In this step, electrostatic, Van Der Waals, solvation and surface area energies were calculated by the HawkDock server (http://cadd.zju.edu.cn/hawkdock/) [107–109]. Nevertheless, the efficacy of this tool is contingent upon the quality of the input structures and the selected force field; inaccuracies may occur if the protein-ligand complex is not adequately optimized [62].

## 2.12. Optimization and cloning of the DNA sequence of vaccine

The Java Codon Adaptation software (http://www.jcat.de/Start.jsp) was applied to standardize the vaccine DNA by making use of the *E. coli* K12 [110]. By determining the GC percentages and the codon adaptation index (CAI) of a protein,

the server evaluated the protein's expression probability [101,110]. Subsequently, the standardized DNA was inserted into the pET-28a(+) vector, with the restriction sites Eco53I and FspI ligated into the vaccine DNA, and cloning was performed in SnapGene (https://www.snapgene.com/free-trial/) [111]. This software is robust for viewing and modifying DNA sequences; nevertheless, it may have constraints for processing exceedingly big or intricate sequences [62].

### 2.13. Immune response simulation

The vaccine-induced immune response simulation was assessed using the C-ImmSim server (https://kraken.iac.rm.cnr.it/C-IMMSIM/) [112,113]. It possesses the ability to simulate the immune response elicited by an epitope that engages with a T cell receptor. The server is capable of simulating three components of the mammalian immune system: the bone marrow, thymus, and lymph nodes. The three administrations of the vaccine were spaced apart by intervals of 1, 84, and 168 time-steps, respectively. However, the three vaccine doses were administered at intervals of 0, 28, and 56 days, corresponding to a real-life time-step equivalence of 8 hours [114,115]. Nevertheless, the simulation parameters were ultimately established as follows: a simulation step of 1000, a simulation volume of 50, and a random seed of 12,345 [62,116].

### 2.14. Modelling of messenger RNA backbones

Initially, the transcription into RNA sequences was carried out utilizing the Biomodel server (http://biomodel.uah.es/en/lab/cybertory/analysis/trans.htm) [71]. Next, the RNA sequence was submitted to the RNAfold server (http://rna.tbi.univie.ac.at/cgi-bin/RNAWebSuite/RNAfold.cgi) for structure modelling [111,117].

## 3. Results

### 3.1. Obtaining of sequence and alignment

The amino acid sequences of aminopeptidase, heat shock protein, P23 protein, serine protease, and sporozoite glycoprotein from both *C. hominis* and *C. parvum* were obtained, and conserved sequences were generated by multiple sequence alignment. These sequences were utilized throughout the study (Fig 1).

### 3.2. Prediction of CD8+ (MHC-I) epitope

The IEDB database was implemented to predict the best CD8+ epitopes of the aminopeptidase, heat shock protein, P23 protein, serine protease and sporozoites glycoprotein. A net of ten 9-mer length epitopes were chosen in accordance with their percentile rank (≤1.00), antigenic (threshold = 0.4), non-allergenic (threshold=<0.00) and non-toxic properties (threshold=<0.00) (Table 1).

### 3.3. Prediction of CD4+ (MHC-II) epitopes

A net of ten 15-mer length epitopes were derived from the targeted proteins based on percentile rank (≤1.00), antigenic (threshold = 0.4), non-allergenic (threshold = 0.5) and non-toxic properties (threshold = 0.5) (Table 2).

### 3.4. Prediction of continuous B-cell epitope

A net of ten epitopes were selected based on antigenic, non-allergenic and non-toxic properties (Table 3).

### 3.5. Analysis of the population coverage of the selected epitope

The vaccine is predicted to have 100% population exposure for both MHC class-I (CD8+) and class-II alleles (CD4+) in regions including the Central America, the South America, the North America, the Oceania, the West Indies, the North Africa, the Central Africa, the West Africa, the East Africa, the Europe, the South Asia, and the Northeast Asia. Following this, Southeast Asia exhibited a coverage of 99.98%, while Southwest Asia reached 99.99%, with global coverage of 100% for combined alleles, 98.55% for MHC class-I alleles, and 99.99% for MHC class-II alleles (S1 Fig).

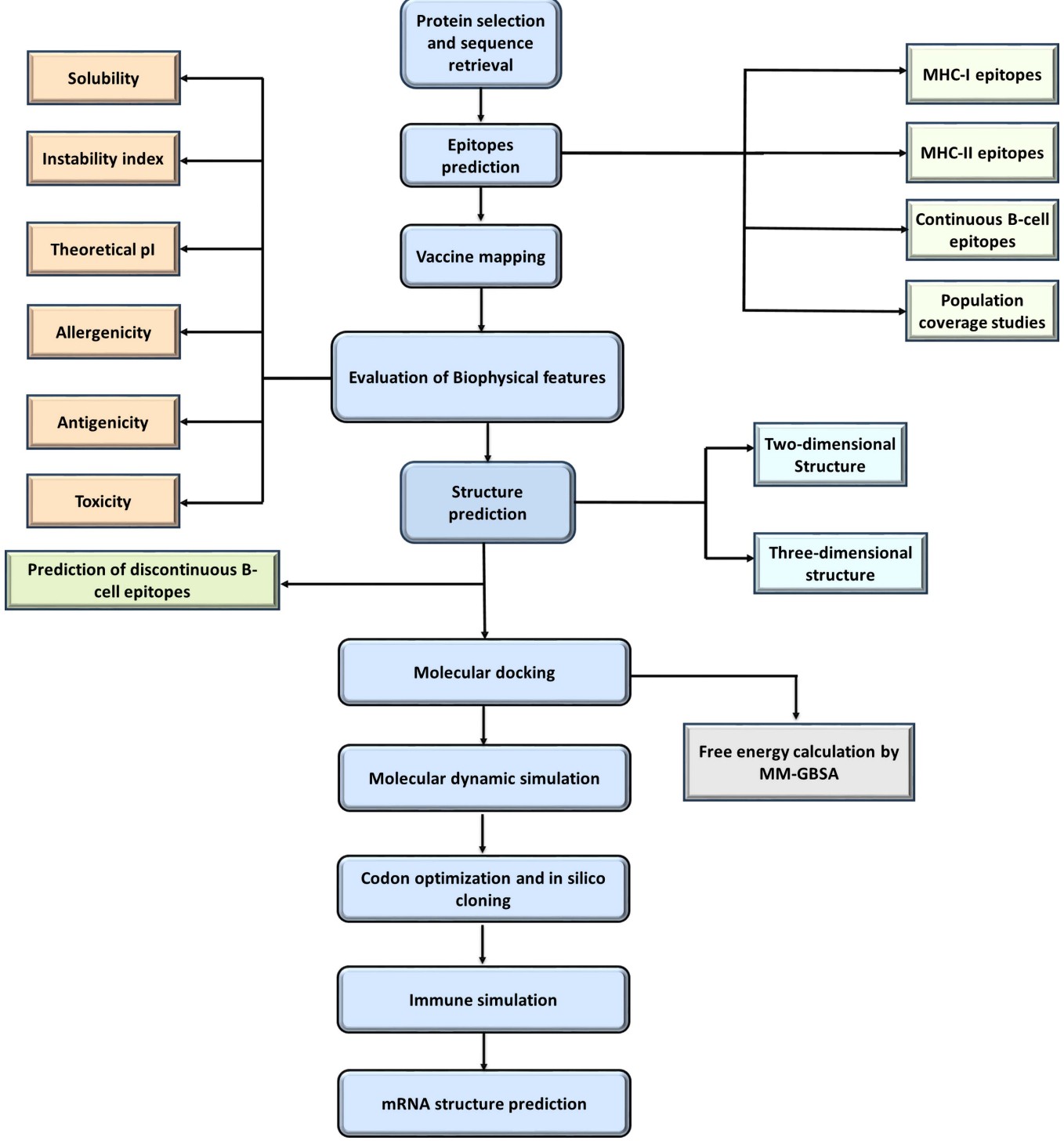

**Fig 1. A brief flow diagram of this study.**

**Table 1. Index of selected CD8+ epitopes of aminopeptidase, heat shock protein, P23 protein, serine protease and sporozoites glycoprotein with their antigenic, non-allergenic and non-toxic properties.**

| Protein | Epitopes | Percentile rank | Allele | Antigenic property | Allergenic property | Toxic property |
|---|---|---|---|---|---|---|
| Aminopeptidase | MSEYTPDKY | 0.01 | 27 | 1.1613 (Yes) | No | No |
| | YLHGTGHGV | 0.03 | 27 | 0.7920 (Yes) | No | No |
| Heat shock protein | ETAGGVMTK | 0.01 | 27 | 0.7542 (Yes) | No | No |
| | STRIPKVQA | 0.02 | 27 | 0.5408 (Yes) | No | No |
| P23 protein | KQRELAEKK | 0.06 | 27 | 0.5884 (Yes) | No | No |
| | APQDKPAEA | 0.19 | 27 | 0.8685 (Yes) | No | No |
| Serine protease | KVFSPYIMK | 0.01 | 27 | 0.6275 (Yes) | No | No |
| | TSEFSIELY | 0.01 | 27 | 1.2405 (Yes) | No | No |
| Sporozoites glycoprotein | YISGEVTSV | 0.01 | 27 | 0.9026 (Yes) | No | No |
| | SANSSSPTK | 0.02 | 27 | 1.2346 (Yes) | No | No |

**Table 2. Index of the best CD4+ epitopes from aminopeptidase, heat shock protein, P23 protein, serine protease and sporozoites glycoprotein with their antigenic, allergenic and toxic properties.**

| Protein | Epitopes | Percentile rank | Antigenic property | Allele | Allergenic property | Toxic property |
|---|---|---|---|---|---|---|
| Aminopeptidase | VDGRYIVEAKKTATP | 0.01 | 0.6207 (Yes) | 27 | No | No |
| | DGRYIVEAKKTATPE | 0.01 | 0.7027 (Yes) | 27 | No | No |
| Heat shock protein | DEAVAYGAAVQAAIL | 0.06 | 0.8882 (Yes) | 27 | No | No |
| | GKFHLDGIPPAPRGV | 0.07 | 0.1842 (Yes) | 27 | No | No |
| P23 protein | EEPKKSEPASNNPPA | 0.45 | 0.9709 (Yes) | 27 | No | No |
| | PEEPKKSEPASNNPP | 0.58 | 0.9709 (Yes) | 27 | No | No |
| Serine protease | DDDMNGYVDDIYGYD | 0.01 | 1.1761 (Yes) | 27 | No | No |
| | YGYDFANNRGSPVDD | 0.02 | 0.6034 (Yes) | 27 | No | No |
| Sporozoites glycoprotein | EYSLVADDKPFYTGA | 0.03 | 0.7963 (Yes) | 27 | No | No |
| | SEYSLVADDKPFYTG | 0.09 | 0.8083 (Yes) | 27 | No | No |

**Table 3. Index of the best continuous B-cell epitopes with their toxic, allergenic and antigenic properties.**

| Protein | Start | Epitopes | Toxic property | Allergenic property | Antigenic property |
|---|---|---|---|---|---|
| Aminopeptidase | 170 | HGIEYAGESSKSKVNK | No | No | 0.5433 (Yes) |
| | 32 | PHMSEYTPDKYKRREF | No | No | 0.7695 (Yes) |
| Heat shock protein | 111 | TIPAKKTQVFTTYADN | No | No | 0.7823 (Yes) |
| | 275 | HNQTAEKDEFEHQQKE | No | No | 0.5348 (Yes) |
| P23 protein | 73 | AQAPPAPAEPAPQDKP | No | No | 0.5928 (Yes) |
| | 50 | QKPEEPKKSEPASNNP | No | No | 1.0427 (Yes) |
| Serine protease | 1101 | EDEGEEKEEGRKRRWE | No | No | 1.0228 (Yes) |
| | 105 | YQEKDSNQEFNSEKAP | No | No | 0.7286 (Yes) |
| Sporozoites glycoprotein | 58 | SGSAGTATESTATTTP | No | No | 1.3749 (Yes) |
| | 51 | SGSQVTPSGSAGTATE | No | No | 1.2368 (Yes) |

### 3.6. Construction of vaccine candidate

The vaccine construct was generated by selecting the most relevant epitopes from the aminopeptidase, heat shock protein, P23 protein, serine protease, and sporozoite glycoprotein. To join the selected adjuvants and epitopes in the designed vaccine, linkers consisting of EAAAK, AYY, AK, and KFER were used to bridge epitopes and enhance vaccine stability (Fig 2).

### 3.7. Validation of biophysical characteristics

The molar mass of the vaccine was 57570.04 Da, and it comprises 523 amino acids (Table 4). The vaccine's isoelectric point was 5.34, indicating a pH below 7. Next, the GRAVY and hydrophobicity scores were −0.829 and −0.345, respectively. Additionally, the solubility overexpression score was 639636. Subsequently, the aliphatic and instability indices were 53.17 and 38, respectively (Table 4). The lack of allergenic reactions to this vaccine was confirmed by the AllerTOP v. 2.0, AllergenFP v. 1.0, and AlgPred servers (Table 4). The vaccine was also recognized by the ANTIGENpro and VaxiJen 2.0 servers as a possible antigen (Table 4).

### 3.8. Prediction of two-dimensional structure

A combination of the GOR4, SOPMA, and PSIPRED servers was deployed to anticipate the two-dimensional structure of the vaccine. As to the GOR4, the percentage of alpha helices was 43.02%, random coils accounted for 46.65%, and extended strands (beta sheet) accounted for 10.33%. Additionally, the SOPMA predicted that the two-dimensional structure of the vaccine would comprise 37.67% alpha helices, 12.62% extended strands, and 42.45% random coils (Fig 3 **and S1 Table**).

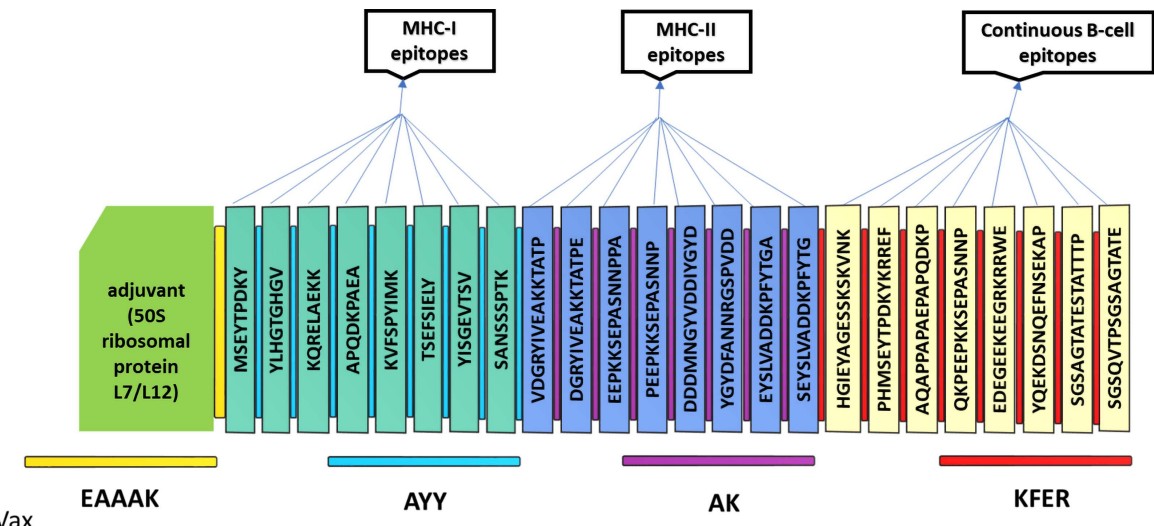

>CrHmVax

MAKLSTDELLDAFKEMTLLELSDFVKKFEETFEVTAAAPVAVAAAGAAPAGAAVEAAEEQSEFDVILEAAAGDKKIGVIKVVREIVSGLGLKEAK
DLVDGAPKPLLEKVAKEAADEAKAKLEAAGATVTVK**EAAAK**MSEYTPDKY**AAY**YLHGTGHGV**AAY**KQRELAEKK**AAY**APQDKPAEA**AAY**KV
FSPYIMK**AAY**TSEFSIELY**AAY**YISGEVTSV**AAY**SANSSSPTK**AAY**VDGRYIVEAAYKTATP**AK**DGRYIVEAKKTATPE**AK**EEPKKSEPASNNPPA**A**
**K**PEEPKKSEPASNNPP**AK**DDDMNGYVDDIYGYD**AK**FSPIYSEAFVPALPS**AK**EYSLVADDKPFYTGA**AK**ISGEVTSVSFEKSES**AK**HGIEYAGES
SKSKVNK**KFER**PHMSEYTPDKYKRREF**KFER**AQAPPAPAEPAPQDKP**KFER**QKPEEPKKSEPASNNP**KFER**EDEGEEKEEGRKRRWE**KFER**Y
QEKDSNQEFNSEKAP**KFER**SGSAGTATESTATTTP**KFER**SGSQVTPSGSAGTATE

**Fig 2. The multiepitope mRNA vaccine construct against *C. hominis* and *C. hominis* with adjuvant (green), epitopes (CD8 + /MHC-1-paste, CD4 + /MHC/2-blue, continuous/linear B-cell-golden), and linkers (EAAAK-yellow, AYY-sky blue, AK-violet, KFER-red).**

**Table 4. Biophysical characteristics of the vaccine.**

| Biophysical characteristics | Scores |
|---|---|
| Molar mass (Da) | 57570.04 |
| Amino acid amount | 523 |
| Isoelectric point | 5.34 |
| GRAVY | −0.829 |
| Instability index | 38 |
| Aliphatic index | 53.17 |
| Net atoms | 8028 |
| Solubility (SOSUI/ SOLpro) | Soluble protein |
| Allergenic property (AllerTOP v. 2.0/ AllergenFP v.1.0/ AlgPred) | No |
| Antigenic property (VaxiJen 2.0/ ANTIGENpro) | Yes |

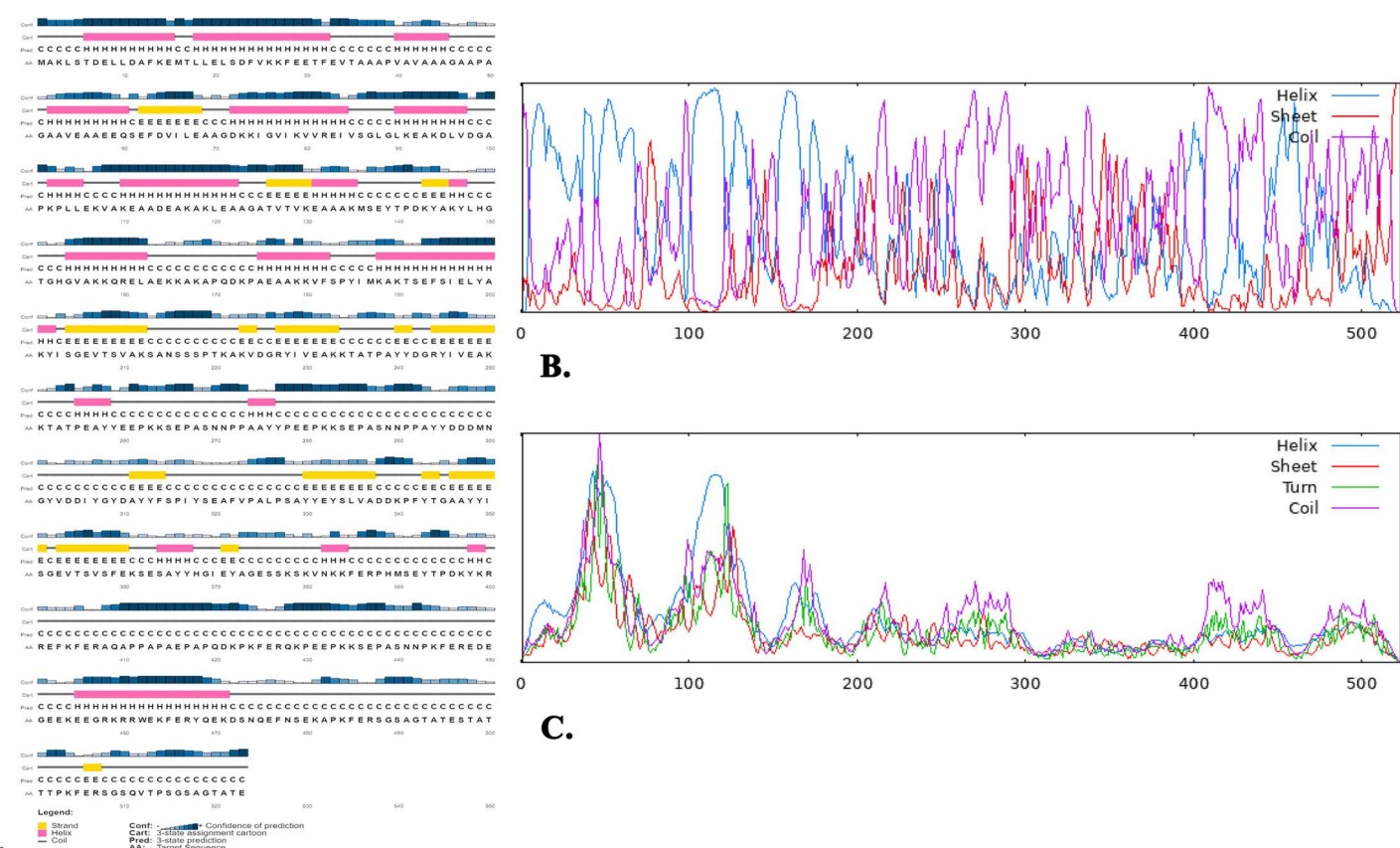

**Fig 3. The PSIPRED generated two-dimensional structure of vaccine (A) where the initial bar (Conf) shows the confidence level in the forecast; different sizes of the bar indicate different degrees of confidence.** Yellow represents the beta-sheet, pink the helix, and grey the vaccine's coil structure in the second bar (Cart). The three separate amino acid sequences and structural aspects are represented by the third (Pred) and fourth (AA) bars, respectively. Additionally, the amount of the helix, sheet, coil and turn by GOR4 (B) and SOPMA (C) servers.

### 3.9. Modelling, refining and validation of three-dimensional structure

From the five structures modelled by the I-TASSER server, the first structure was chosen for a superior template modelling value of 0.65±0.13, a confidence value of −0.52, and an RMSD of 8.6±4.5 Å (Fig 4A). Subsequently, the refined three-dimensional structure was obtained from the GalaxyWEB server (Fig 4B). The stability of the three-dimensional structure was supported by a Ramachandran favoured region of 90.6%, an RMSD value of 0.547, and a MolProbity score of 1.973, all of which reflect its high quality. This result confirmed that the refined model is appropriate for subsequent

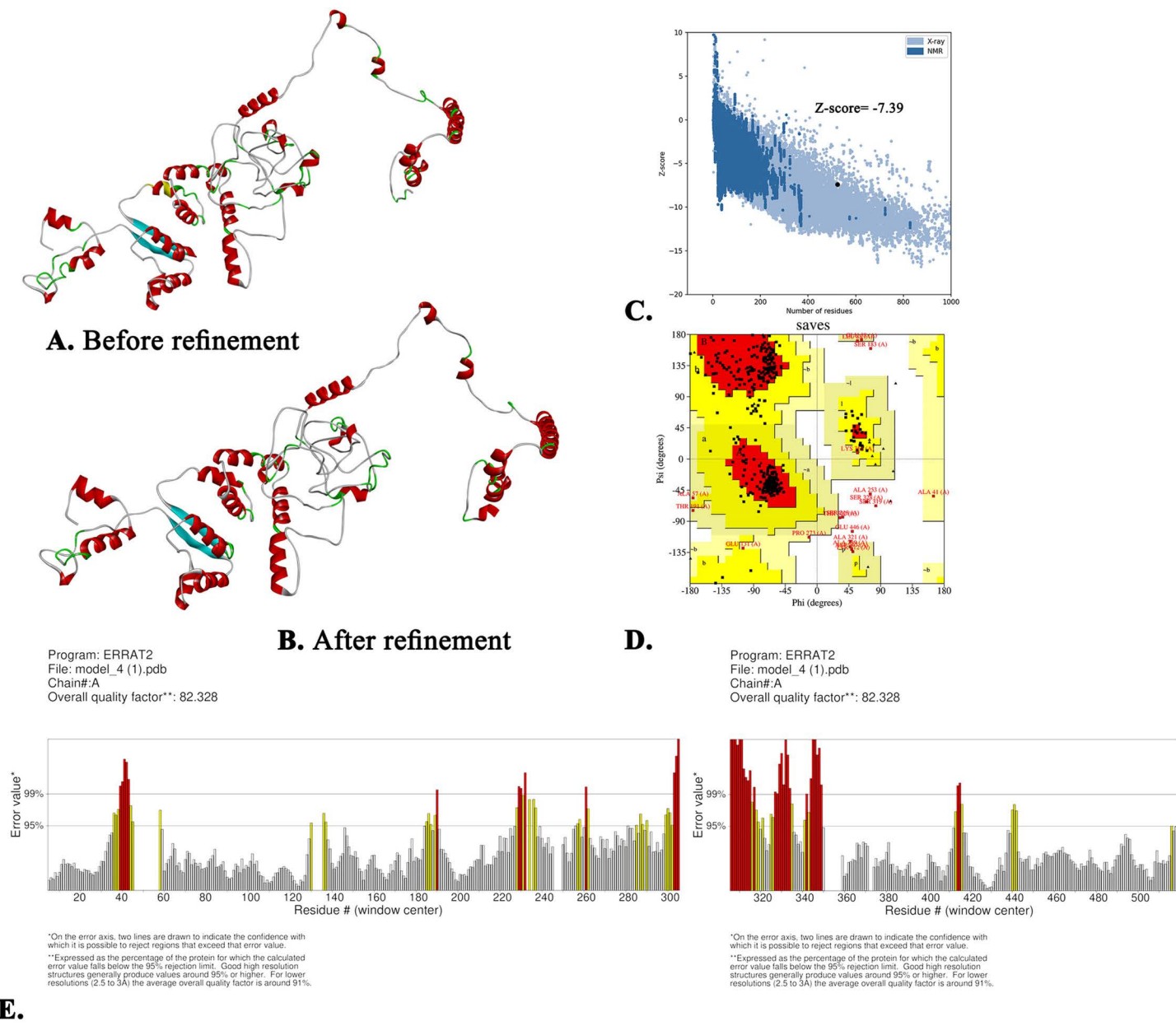

**Fig 4. Prediction and validation of the predicted three-dimensional model of the vaccine with (A) non-refined structure (B) refined structure, (C). Z-score, (D) Ramachandran plot scores and (E) ERRAT score.**

analysis. Next, the refined structure obtained a Z-score of −7.39, which signifies its high quality (Fig 4C). Conversely, the Ramachandran plot showed that the refined structure placed 83.1% of the amino acid residues in the most favoured regions, 13% in the additionally allowed regions, and 2.2% in the generously allowed regions (Fig 4D). Additionally, this vaccine also exhibited an ERRAT score of 82.328, as confirmed by the SAVES server (Fig 4E).

### 3.10. Prediction of discontinuous B-cell epitopes

From a net of 621 amino acids in the vaccine, the Discotope 2.0 server detected 259 discontinuous B-cell epitopes ranging from 0.528 to 0.993 (Fig 5 **and** S2 Table).

### 3.11. Docking interaction analysis

Using the ClusPro 2.0 server, the coupling procedure between the vaccine construct and human receptors (TLR-2 and TLR-4) was executed. However, ClusPro 2.0 generated 40 models for each complex. Among the models considered, the one with the lowest pose energy of −1151.9 and the central pose energy of −1151.9 was selected for the vaccine-TLR-2 complex (Table 5). Subsequently, the model with the lowest pose energy value of −1028.3 and the central pose energy value of −902.6 kJ/mol was selected for the vaccine-TLR-4 complex (Fig 6 **and** Table 5). Next, the vaccine-TLR-2 complex contained 2, 2, 120 salt bridges, hydrogen bonds, and non-bond contacts, respectively (Table 5). Subsequently, the vaccine-TLR-4 complex contained 1, 23, 225 salt bridges, hydrogen bonds, and non-bond contacts, respectively (Table 5).

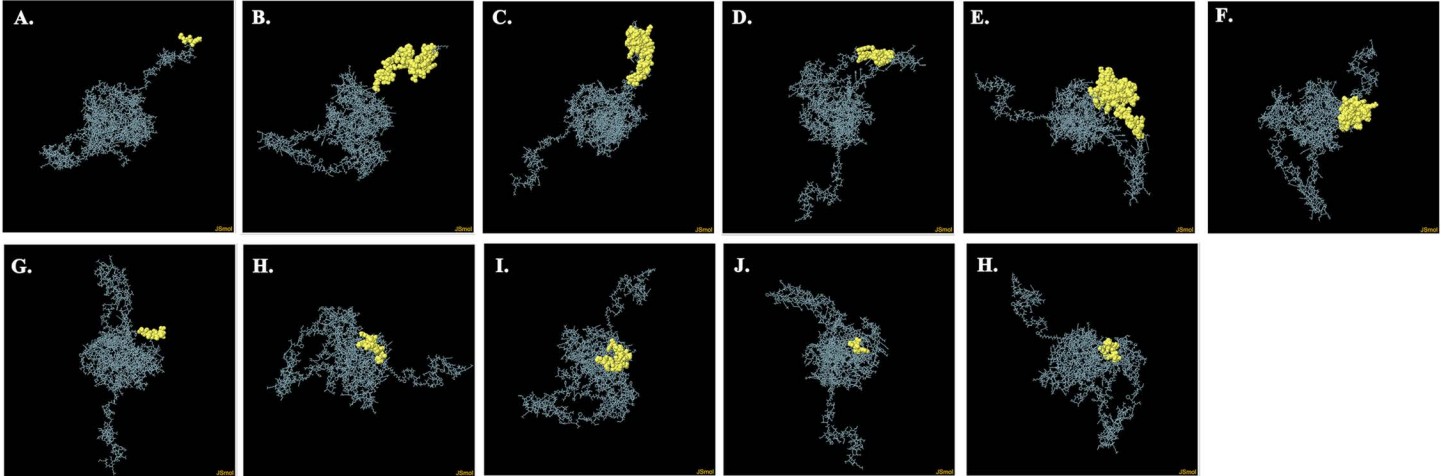

**Fig 5. Three-dimensional illustration of discontinuous B-cell epitopes of the vaccine (A–K).** Yellow-color showcases the discontinuous B cell epitopes and the grey sticks represents the vaccine.

**Table 5. The energy values and bonding features between the ligand-receptor.**

| Ligand-Receptor | Energy values | | Bonding features between ligand and receptor | | | |
|---|---|---|---|---|---|---|
| | Central pose | Lowest pose | Salt bridges | Disulfide bonds | Hydrogen bonds | Non-bonded contacts |
| **Vaccine-TLR-2** | −1151.9 | −1151.9 | 2 | – | 2 | 120 |
| **Vaccine-TLR-4** | −902.6 | −1028.3 | 1 | – | 23 | 225 |

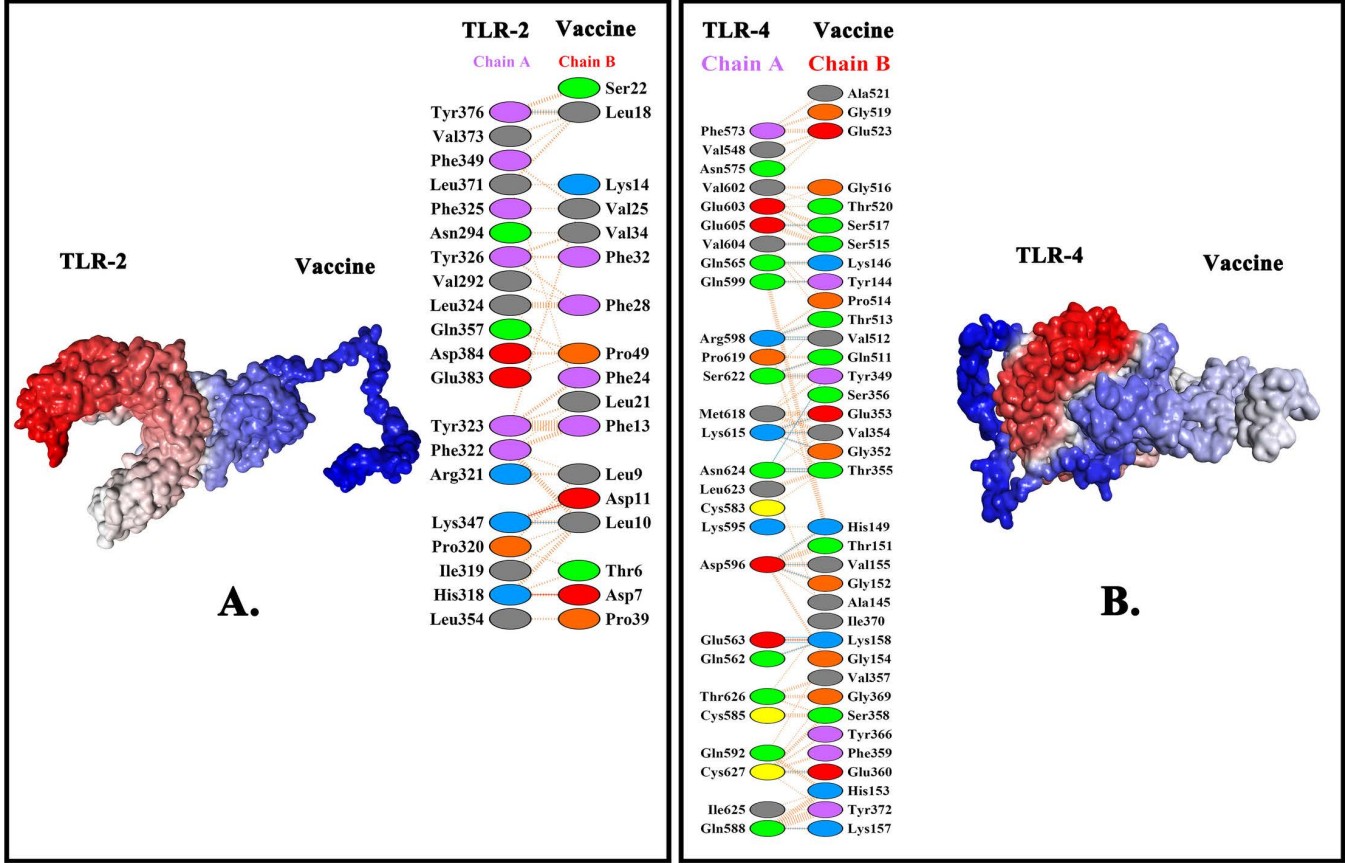

**Fig 6. The interaction between the vaccine construct and TLR-2 (A) and TLR-4 (B).**

### 3.12. MM-GBSA analysis

In the case of vaccine-TLR-2, the total negative energy (MM-GBSA) was −63.3 kcal/mol, while the Van der Waals, electrostatic, generalized born, and surface area energies were −125.07 kcal/mol, 528.6 kcal/mol, −451.22 kcal/mol, and −15.61 kcal/mol, respectively. For the vaccine-TLR-4, the total negative energy was calculated to be −70.17 kcal/mol, while Van der Waals, electrostatic, generalized born, and surface area energies were −187.78 kcal/mol, 1082.03 kcal/mol, −940.48 kcal/mol, and −23.94 kcal/mol, respectively (S3 Table).

### 3.13. Dynamic simulation of the molecules

To evaluate structural integrity and modifications, the iMODS server conducted molecular dynamics simulations of the vaccine-bound complexes with TLR-2 and TLR-4. The covariance maps generated during the analysis of molecular dynamics simulations depicted the correlated, uncorrelated, and anti-correlated shifts between residue pairs in the docked complexes of the vaccine with TLR-2 and TLR-4, utilizing red, white, and blue hues to represent these relationships, respectively (Figs 7A and 7B). Subsequently, the elastic map of the docked complexes showed interactions between atoms as darker grey patches, suggesting more rigid regions (Figs 7C and 7D). However, an eigenvalue measures the impact of specific deformation motions on the protein's overall dynamics. Additionally, an elevated eigenvalue indicates significant displacement, whereas a reduced eigenvalue indicates collective conformational changes in the protein complex.

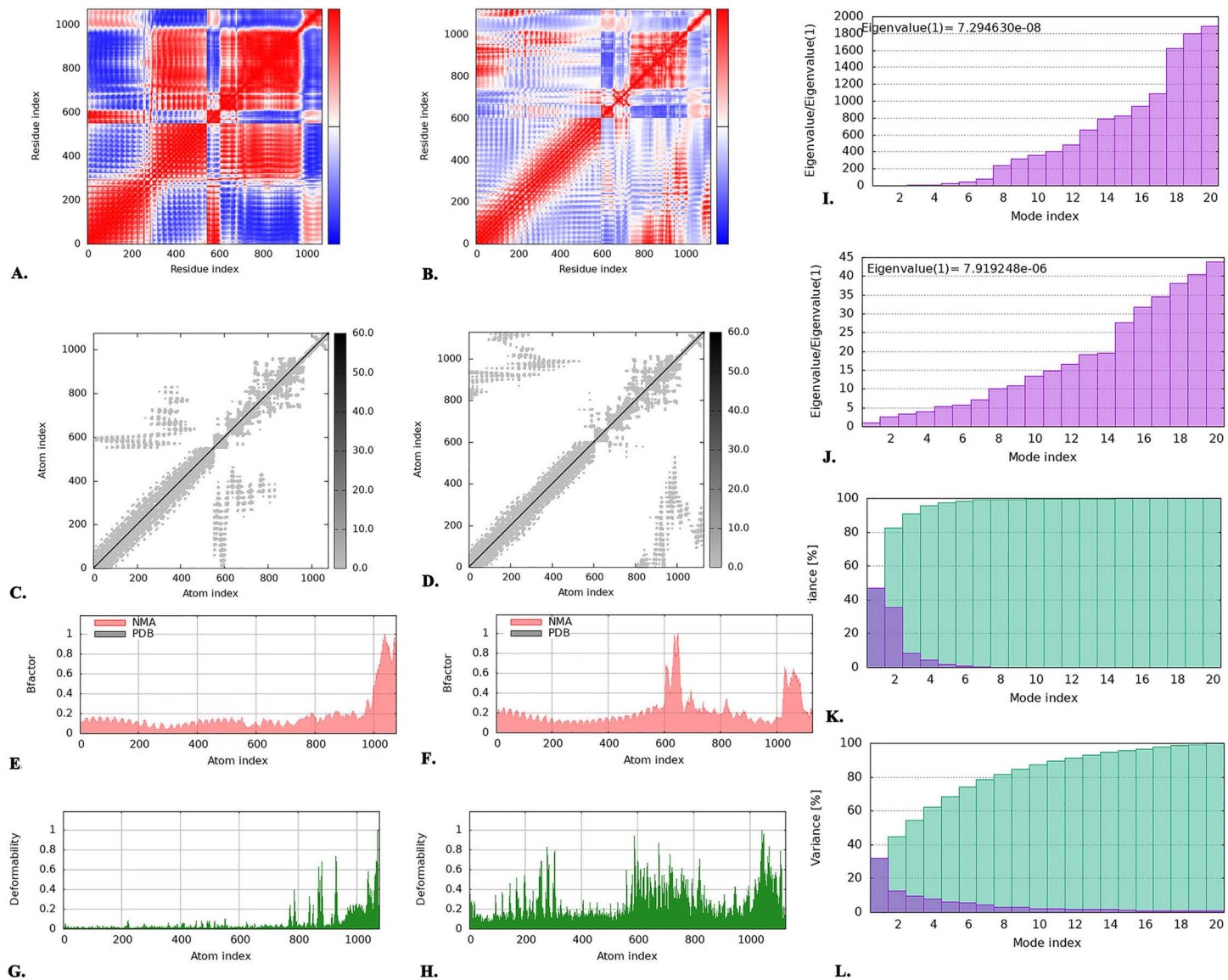

**Fig 7. The analysis of the dynamic simulation of molecules for both vaccine-TLR-2 and vaccine-TLR-4 complexes.** A representation of **(A)** covariance graph (correlated, uncorrelated, and anti-correlated motions are represented by red, white, and blue regions, respectively, **(C)** network of elasticity, **(E)** B factor, **(G)** deformability, **(I)** eigenvalue, and **(K)** the variance of vaccine-TLR-2 complex. There exists another representation of **(B)** covariance graph (correlated, uncorrelated, and anti-correlated motions are represented by red, white, and blue regions, respectively, **(D)** network of elasticity, **(F)** B factor, **(H)** deformability, **(J)** eigenvalue, **(L)** variance of vaccine-TLR-4 complex.

Next, the B-factor graph showed simulations of the docked complexes involving NMA and the PDB sector, where B-factor values indicated the amplitude of atomic displacements, revealing higher deformability in the vaccine-TLR-2 and vaccine-TLR-4 docked complexes, suggesting greater flexibility (Figs 7E **and** 7F). Using graphical peaks, the deformability graph showed that the flexible portions of the docked complexes were more deformable in vaccine-TLR-4 than in vaccine-TLR-2 (Figs 7G **and** 7H). Furthermore, the eigenvalues of the vaccine-TLR-2 and vaccine-TLR-4 docked complexes indicated their structural stability, with the lowest eigenvalues of 7.294630e-08 and 7.919248e-06, respectively (Figs 7I **and** 7J). The amplitude of each mode's fluctuation in normal mode analysis

(NMA) is inversely related to the mode's eigenvalue, which is used to assess molecular flexibility, and this relationship is statistically represented by the variance. The variance graph illustrated the cumulative variance in cyan and the individual variance in purple. In the vaccine-TLR-2 complex, the initial two modes, out of a total of twenty, account for 80% of the observed variance, whereas in the vaccine-TLR-4 complex, the first seven modes contribute 80% of the observed variance (Figs 7K and 7L).

### 3.14. Optimization and cloning of the DNA sequence of vaccine

The Java Codon Adaptation software was employed to optimize codon usage in *E. coli* K12, thereby enhancing protein expression for vaccine development. However, the server predicted that a codon sequence of 981 nucleotides was optimized. Additionally, the overall GC percentage of 47.02% and the codon adaptation index (CAI) of 0.9664 for the adapted sequence indicated that the protein would exhibit elevated expression. However, the cloning was completed by inserting the adapted codon sequences into the plasmid vector pET-28a(+) using SnapGene software (Fig 8).

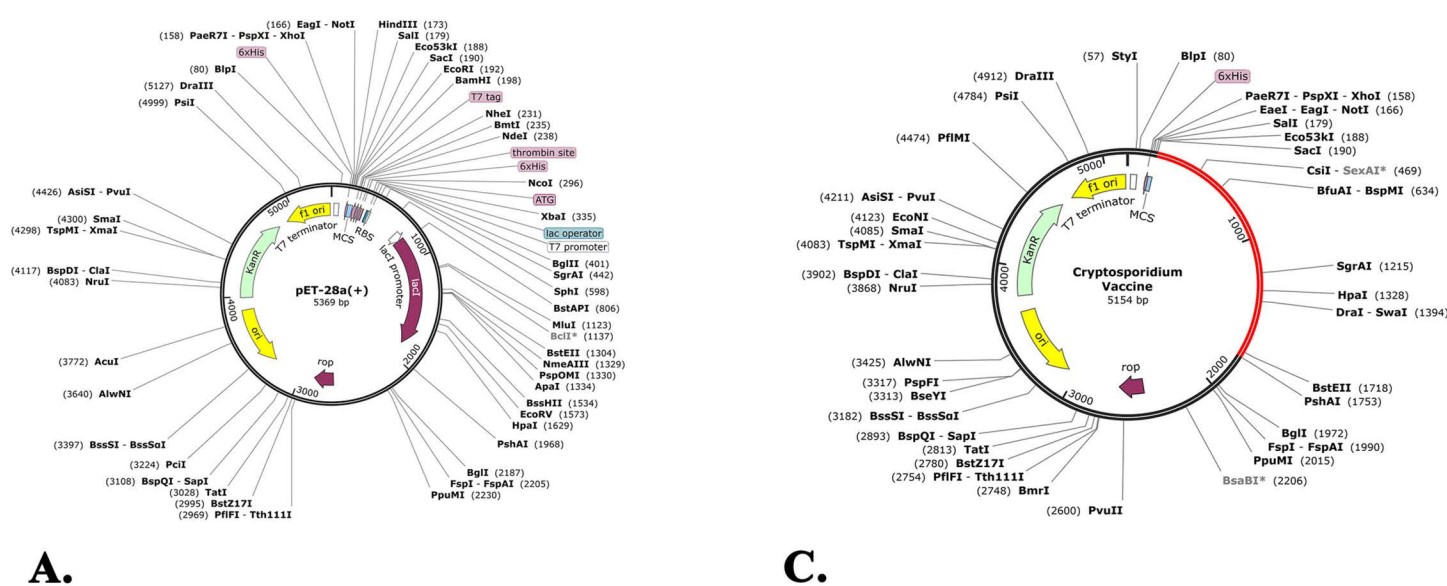

**A.**

**C.**

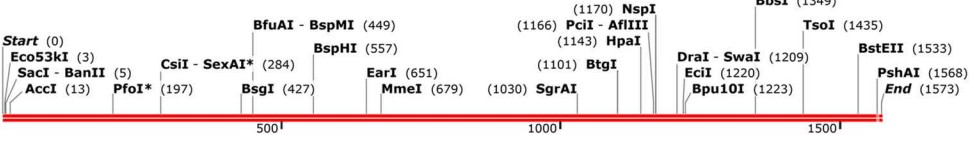

**B.**

**Cryptosporidium hominis DNA**
1573 bp

**Fig 8. Cloning of the vaccine construct.** The depiction was about the **(A)** vector, **(B)** DNA of sequence of the vaccine and **(C)** cloned construct of the vaccine.

### 3.15. Immune response simulation

The B-cell population in the vaccine group increased significantly and stayed active after the triple-dose regimen, suggesting prolonged protection (Fig 9A). T-cell responses, including both TH and TC cells, were enhanced by the vaccine. Active TH cell levels peaked on day 50, then gradually declined, persisting for almost a year (Fig 9B). Active and resting TR cells rose after the initial dose, declined after day one, yet remained present for a year (Fig 9C). Active TC levels increased after vaccination and sustained activity for a year (S2 Figs C and D). In contrast, NK, DC, and MA cell levels remained steady during the simulation (Figs 9D-9F). IgM+IgG concentrations steadily increased, peaking after vaccination (S2 Fig A). Lastly, the vaccine activated several cytokines, notably in the IFN and IL-2 families (S2 Fig B).

### 3.16. Modelling of messenger RNA backbones

The mRNA vaccine had an MFE score of −388.70 kcal/mol (optimal structure) and −250.50 kcal/mol (centroid structure). Subsequently, the predicted thermodynamic free energy was −414.35 kcal/mol. Additionally, the vaccine's MFE structures were associated with a frequency of 0.00% in the ensemble (Fig 10).

## 4. Discussion

The significance of vaccination's influence on the well-being of the global population is difficult to overstate. Aside from potable water, no other method has had such a significant impact on reducing mortality rates and increasing population growth [118]. In recent years, RV and immunoinformatics have transformed vaccine research into a cutting-edge approach by identifying optimal epitopes without growing pathogens in the laboratory [119].

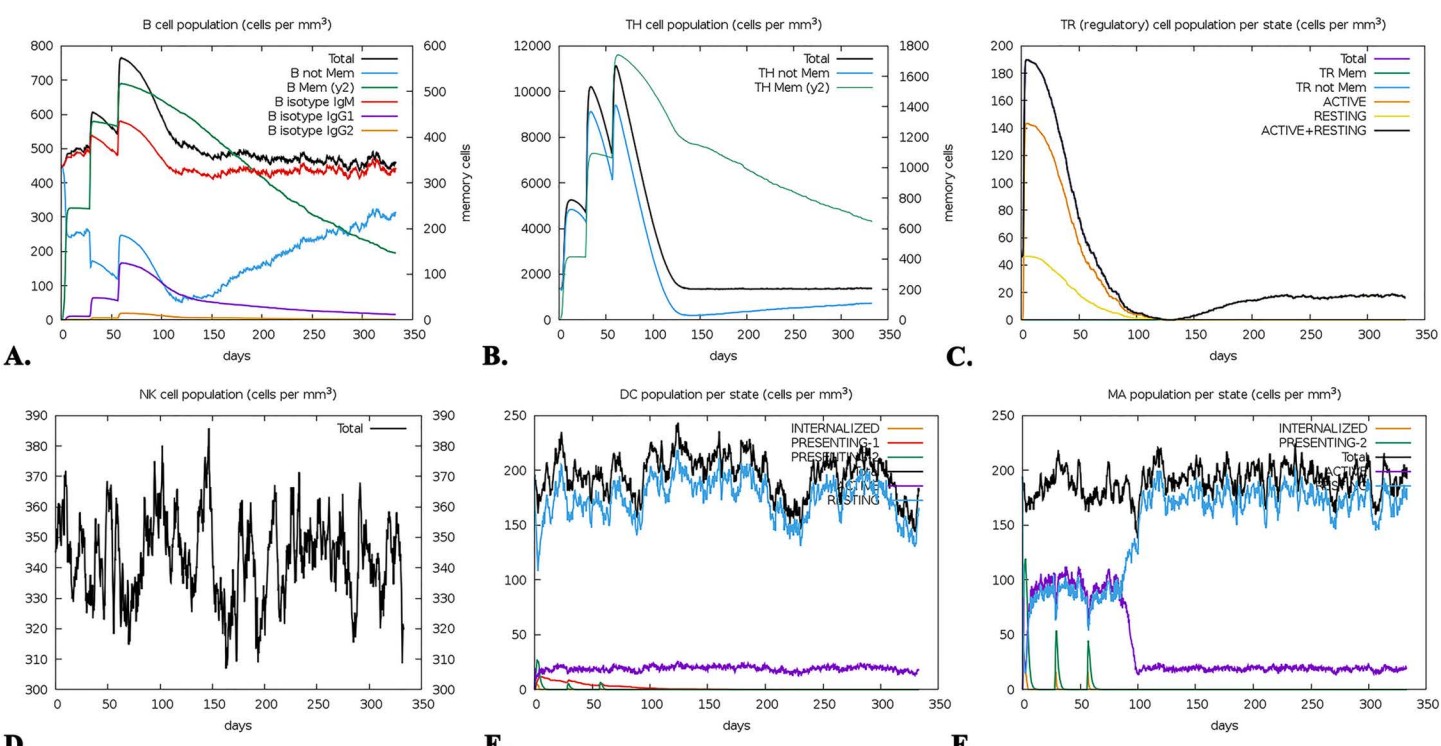

**Fig 9. The immune simulation of the vaccine, where the illustration is depicted as (A) The lifespan of B-cell, (B) TH-cell, (C) TR-cell, (D) NK cell, (E) DC cell and (F) MA cell.**

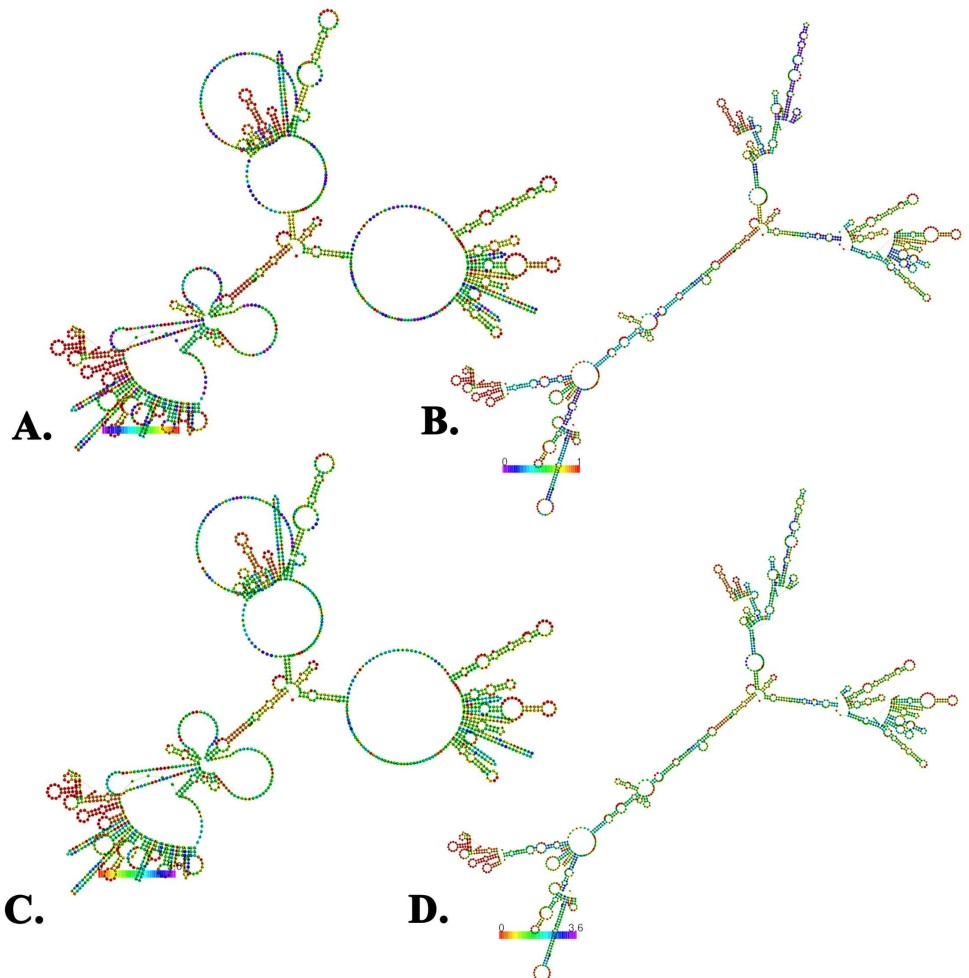

**Fig 10. The predicted mRNA structures of the vaccine by the RNAfold web server.** The centroid structures of the vaccine with the base pair probabilities **(A and B)** and the positional entropy **(C and D)**.

Cryptosporidiosis, the primary cause of endemic and pandemic diarrheal disease worldwide, is increasingly identified in immunosuppressed Americans [120]. Persistent gastroenteritis is the primary cause of mortality among children under the age of five in developing nations, contributing to 30–50 percent of such fatalities [121]. Additionally, *C. parvum* causes gastrointestinal disease in juvenile ruminant livestock, with neonatal calves most affected [122]. Currently, the absence of vaccines and the impracticality of chemotherapeutics make immunoprophylactic agents and treatments for cryptosporidiosis critical concerns [123]. The primary aim of this study was to construct a multiepitope mRNA vaccine against *C. hominis* and *C. parvum* using RV and immunoinformatics techniques.

However, only three RV-based vaccines have been specifically designed against cryptosporidiosis. In these three studies, (1) Sethi, G., et al., (2025) targeted Cp15, Cp23, and CpP2 proteins, (2) Karimzadeh, F., et al., (2025) focused on some experimentally identified epitopes of *Cryptosporidium* spp, and (3) Dhal, Ajit Kumar., et al., (2019) designed the vaccine by targeting seven proteins out of thirty-eight proteins of the parasite [124–126].

In our study, five conserved proteins (aminopeptidase, heat shock, P23, serine protease, and sporozoite glycoproteins) were targeted of the *C. hominis* and *C. parvum* to predict the best CD8+, CD4+, and continuous B-cell epitopes based on

antigenicity, allergenicity, toxicity, and percentile rank. As five of the vital proteins were targeted specifically along with dual species of parasites, this current study is novel and superior to the previously mentioned studies. To initiate host immune responses against intracellular infections and virus-infected cells, CD8+ epitopes are essential. To activate their response, CD8+ cells attach to MHC-I molecules. Therefore, intracellular infections can be effectively eradicated by MHC-I epitopes of vaccines, which can significantly activate CD8+T cells [127]. Furthermore, CD4+ epitopes are essential for delivering peptides to the surface of CD4+T cells, enabling interaction with T-cell receptors (TCR) and driving both antibody-mediated and cellular immune responses. Moreover, the TCR and MHC-II molecules are instrumental in preventing microbial infections, rejecting transplants, and monitoring cancer progression [104,128–131]. Therefore, the recognition of CD4+ and CD8+T-cells is a prerequisite for the use of mRNA vaccines [132–134]. Furthermore, the critical function that B-cell epitopes play in the interaction between antigens and antibodies has been widely recognized as a critical element of vaccine development [135–137]. Consequently, the combined CD8+ and CD4+ epitopes exhibited high global coverage across Oceania, the United States, Asia, Europe, the West Indies, and Africa, indicating that our vaccine will likely encompass a larger segment of the global population. However, the population coverage was conducted only on the (1) mentioned study where they also found good coverages of combined MHC-I and MHC-II epitopes [126]. But the mentioned (2) and (3) studies are lack of population coverages so the current study is superior to those studies [124,125].

For the vaccine construction, all the mentioned (1), (2) and (3) utilized identical linkers including EAAAK, AYY etc which provide high solubility and stability but all of those studies utilized different adjuvants compared to this current study (large ribosomal subunit protein of *Mycobacterium tuberculosis*) which provides high immunogenicity [100,101,111,124–126].

Based on biophysical features, this vaccine was soluble; thus, its functional stability in the human system was apparent [132,138]. The vaccine exhibited significant heat stability, as evidenced by an aliphaticity index of 53.30 [104,132,139]. The vaccine's hydrophilic character was indicated by a GRAVY score of −0.829 where the (1), (2) and (3) mentioned studies had the GRAVY scores of −0.106, −0.492 and 0.091 meaning that the vaccine of this current study is more hydrophilic and soluble compared to others [94,124–126]. Subsequently, both the two-dimensional and three-dimensional structures of the vaccine were modeled to assess their significance in protein folding [71]. The vaccine's two-dimensional structure comprises a considerable quantity of alpha helices (43.02%) and coils (10.33%), rendering it a highly stable protein. However, the mentioned (1) study has better quantity of alpha helices (64.67%) and (32%) compared to this current study [126]. Furthermore, the Ramachandran plot assessment demonstrated that a substantial amount (83.1%) of the amino acid residues remained in the most favored areas, thereby ensuring the validity of its three-dimensional structure. Notably, the mentioned (1), (2) and (3) studies had better Ramachandran plot outcome as 93.8%, 85.7% and 89.2% in the most favored regions, indicating better structural features of the vaccine compared to the current study [124–126]. The Z-score of −7.39 further indicates the overall good quality (negative means high quality) of the vaccine model. However, the mentioned (1) and (2) had Z scores of −8.83 and −1.88, respectively, indicating that the model of (1) study had better quality [125,126]. In addition, the ERRAT score of 82.328 was determined to be indicative of a high-quality structure, as scores >50 indicate excellent quality [70]. Next, the human TLR-2 and TLR-4 were used in a docking analysis to assess the binding affinity between the vaccine and TLRs expressed on immune cells. Nevertheless, the vaccine exhibited notable binding affinity for TLR-2 (lowest energy of −1151.9) and TLR-4 (lowest energy of −1028.3). However, in the (1), (2) and (3) studies, the docking scores were −1328.5 kcal/mol, −214.8 and −966.9 kcal/mol, respectively indicating the better interactions of (1) and (3) compared to the current study [124–126]. The docking result was validated by the MM-GBSA scores of −63.3 kcal/mol (vaccine-TLR-2) and −70.17 kcal/mol (vaccine-TLR-4). Thereafter, the iMODs server conducted dynamics simulations of the molecules for docked complexes of the vaccine-TLR-2 and vaccine-TLR-4. An eigenvalue measures the influence of certain deformation movements on the protein's overall motion. A higher eigenvalue signifies significant displacement, whereas a lower eigenvalue is associated with collective conformational alterations in the protein complex [140]. Additionally, the docked complexes of vaccine-TLR-2 and vaccine-TLR-4 demonstrated the lowest eigenvalues of 7.294630e-08 and 7.919248e-06, respectively. The results indicated that both complexes were biologically

stable. However, the amplitude of fluctuations for each mode in normal mode analysis (NMA) has an inverse relationship with the eigenvalue, a relationship crucial for evaluating molecular flexibility and expressed numerically as variance. In the vaccine-TLR-2, the initial 2 modes account for 80% of the whole variation, but in the vaccine-TLR-4, the first 8 modes account for 80% of the total variance. Additionally, the deformability of a protein complex is crucial for achieving structural compatibility with substrates while maintaining its functional integrity, allosteric regulation, and protein-protein interactions [141,142]. Nonetheless, the vaccine-TLR-4 exhibited greater deformability than vaccine-TLR-2, indicating that vaccine-TLR-4 is functionally more stable. Subsequently, the B factor is a metric that measures the deviation of atoms from their average position, and it serves as a crucial indicator of the region's activity, thermal stability, solvent accessibility, and protein flexibility [143]. The B-factor scores indicate the extent of atomic displacement and indicate that vaccine-TLR-2 and vaccine-TLR-4 demonstrated greater deformability and flexibility. However, the vaccine-TLR-2 had Van der Waals, electrostatic, generalized born, and surface area energies of −125.07 kcal/mol, 528.6 kcal/mol, −451.22 kcal/mol, and −15.61 kcal/mol, respectively. For the vaccine-TLR-4, the Van der Waals, electrostatic, generalized born, and surface area energies were −187.78 kcal/mol, 1082.03 kcal/mol, −940.48 kcal/mol, and −23.94 kcal/mol, respectively. All of the energy scores of MM-GBSA were strong because <−80 (Van der Waals), more negative value (electrostatic), (<−200) and <−5–10 (surface area energies) benchmarks indicate strong binding interactions [106].

Consequently, the vaccine's DNA sequence was optimized to assess the likelihood of expression in the vector. The results demonstrated that the vaccine showed high expression in the vector, as evidenced by a GC content of 47.02% and a CAI of 0.9664. Consequently, the vaccine has shown the capacity to provoke both innate and adaptive immune responses, evidenced by the detectable presence of continuous B cells and memory B and T cells for one year. The initial rise in IFN-γ and IL-2 levels after the initial injection, and their sustained levels with recurrent antigen exposure, were evident features of TH cell activation and the subsequent release of IFN-γ and IL-2. In addition, the data suggested the existence of a humoral immune response, as evidenced by the synthesis of antibodies (IgM and IgG). However, this current vaccine was superior to the mentioned (1), (2) and (3) studies as it could elicit higher immune cells compared to theirs [124–126]. At last, the lowest free energy of the mRNA structure of the vaccine was −388.70 kcal/mol for the optimal structure and −250.00 kcal/mol for the centroid structure, thus demonstrating the stability of the vaccine upon entry, transcription, and expression in the host. However, the mentioned (1) study had the free energy values of −430.30 kcal/mol for MFE and −312.42 kcal/mol for centroid structures, respectively indicating better outcome compared to the current study. However, this study was mainly focused on computational prediction, which has a number of limitations. These drawbacks include a lack of long-term immunity prediction, structural prediction errors, ambiguity about adjuvant interactions, inconsistent epitopes, unstable proteins, unintended allergenicity and toxicity etc. As a result, validation of this work requires both preclinical and clinical trials with long term precise inspection by the regulatory bodies such as the Food and Drug Administrations of USA, Centers for Disease Control and Prevention etc.

## 5. Conclusion

The development of innovative vaccines has been substantially advanced by advances in immunoinformatics and RV, which have also improved disease prevention strategies. A novel mRNA vaccine for cryptosporidiosis is introduced in this study, demonstrating the potential of an *in-silico* strategy for selecting and transcribing antigens into mRNA. This vaccine was specifically designed from the best epitopes of the two prevalent parasites associated with this infection, thereby demonstrating high population coverage worldwide and optimal biophysical characteristics. Additionally, docking analyses demonstrated robust interactions between TLR-2 and TLR-4 receptors, findings validated by MM-GBSA and dynamic simulation analyses. The vaccine was potential in activating various immune cells (T and B cells), cytokines and antibodies. Finally, based on the mRNA structure prediction, this vaccine is likely to maintain sufficient stability upon entry, transcription, and expression in the host. These findings provide invaluable insights into the attributes and uses of an effective computationally designed mRNA vaccine for the treatment of cryptosporidiosis. Nevertheless, this study focused solely on

computational prediction, which has some limitations, including a lack of long-term immunity prediction, structural prediction errors, and uncertainty about adjuvant interactions, and thus requires preclinical and clinical trials for validation.

## Supporting information

**S1 Fig. The population coverage of the identified epitopes (MHC-I, MHC-II and combined) that met the criterion for inclusion in the vaccine design.**
(TIF)

**S2 Fig. The simulation of the immune response for vaccine where the illustration is depicted as (A) The antigen and antibody titer count, (B) Cytokine level count, (C) TC cell population and (D) TC cell population per state.**
(TIF)

**S1 Table. The two-dimensional structures of the vaccine.**
(DOCX)

**S2 Table. The predicted discontinuous B-cell epitopes for vaccine.**
(DOCX)

**S3 Table. The energy scores of MM-GBSA for the complexes.**
(DOCX)

## Author contributions

**Conceptualization:** Ahmad Abdullah Mahdeen, Imam Hossain.

**Data curation:** Ahmad Abdullah Mahdeen, Imam Hossain, Md. Habib Ullah Masum.

**Formal analysis:** Ahmad Abdullah Mahdeen, Imam Hossain.

**Investigation:** Imam Hossain, Md. Habib Ullah Masum.

**Methodology:** Ahmad Abdullah Mahdeen, Imam Hossain, T. M. Fazla Rabbi.

**Software:** Ahmad Abdullah Mahdeen.

**Supervision:** Imam Hossain.

**Validation:** Ahmad Abdullah Mahdeen, Md. Habib Ullah Masum.

**Visualization:** Ahmad Abdullah Mahdeen, Md. Habib Ullah Masum.

**Writing – original draft:** Ahmad Abdullah Mahdeen, Imam Hossain, T. M. Fazla Rabbi, Sajedul Islam.

**Writing – review & editing:** Ahmad Abdullah Mahdeen, Imam Hossain, T. M. Fazla Rabbi.

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
