## [Decision Letter · Decision Letter 0]

30 Oct 2025

Dear Dr. Hossain,

Thank you for submitting your manuscript to PLOS ONE. After careful consideration, we feel that it has merit but does not fully meet PLOS ONE’s publication criteria as it currently stands. Therefore, we invite you to submit a revised version of the manuscript that addresses the points raised during the review process.

We look forward to receiving your revised manuscript.

Kind regards,

Shahina Akter, Ph.D.

Academic Editor

PLOS ONE

Journal Requirements:

2. Please remove all personal information, ensure that the data shared are in accordance with participant consent, and re-upload a fully anonymized data set.

Additional Editor Comments:

Dear Authors,

We have reached a decision on your manuscript. While one reviewer recommended rejection without detailed comments, the other reviewers have provided constructive feedback for minor and major revisions. Given the novel premise and overall potential of your work, we invite a major revision. Please address all points raised by Reviewers 2 and 3 in full.

Reviewers' comments:

Reviewer's Responses to Questions

**Comments to the Author**

1. Is the manuscript technically sound, and do the data support the conclusions?

Reviewer #1: No

Reviewer #2: Yes

Reviewer #3: Yes

2. Has the statistical analysis been performed appropriately and rigorously?

Reviewer #1: No

Reviewer #2: Yes

Reviewer #3: Yes

3. Have the authors made all data underlying the findings in their manuscript fully available?

Reviewer #1: No

Reviewer #2: Yes

Reviewer #3: Yes

4. Is the manuscript presented in an intelligible fashion and written in standard English?

Reviewer #1: Yes

Reviewer #2: Yes

Reviewer #3: Yes

Reviewer #1: Biological experiments should be conducted to develop the vaccine. This is a bioinformatic manuscript for vaccine development (PONE-D-25-20787). The language and writing are fine. However, this manuscript is not of great importance to developing the Cryptosporidium parvum vaccine.

Reviewer #2: This is a well written manucript,however some fo the processes in the methodology could be further summerized The The study aim was well stated in lines 148-150 but `` intergrating vaccines`` shouldnt be part of the aim.it fits best inthe introduction section.It would be nice to strucure the abstract,The conclusion as sttted inthe anstract is not apprporroate it should be revised and sated inline withthe study aim.

While the Authours have tried to state the conclusion from theor study ,they have also included some discusion of their conclusion.The conclusion shoud revised and clearly stated in line with the study aim and leave out the discussion .

Reviewer #3: The study presents a computational design of an mRNA-based multiepitope vaccine targeting Cryptosporidium hominis and C. parvum. While the topic is relevant and aligns with the current trend of using immuno-informatics for vaccine development, the manuscript lacks sufficient methodological rigor, biological validation, and critical interpretation of results. Several sections read like a procedural description rather than a scientific analysis, and there are substantial concerns regarding novelty, clarity, data reproducibility, and presentation.

Hence, major revision is strongly recommended before this manuscript can be considered for publication.

While the topic is timely and the approach relevant, the manuscript currently lacks methodological rigor, interpretative depth, and structural clarity. The claims of strong immunogenicity and universal coverage are unsupported by robust comparative analysis or experimental evidence. A comprehensive revision with clear validation, data justification, and scientific moderation is required.

Major Issues That Must Be Addressed

1. Scientific validation: Add benchmarking or comparison to previously published vaccine constructs or experimental epitopes.

2. Novelty and contribution: Clarify what this study adds beyond existing in-silico works on Cryptosporidium.

3. Data reproducibility: Provide detailed parameters, thresholds, and URLs for all tools, including dataset accession IDs.

4. Figures and tables: Improve quality, labeling, and consistency with captions and in-text references.

5. Discussion: Critically interpret findings and discuss biological limitations of in-silico work.

6. Language: Substantial language polishing and reorganization needed to improve readability.

7. Ethical and data statements: Ensure that the data availability and ethical statements follow journal standards and are placed at the end of the manuscript.

My detailed Comments are given as follows

1. Title and Abstract

• The title is clear but overly long. Consider shortening to emphasize the main objective (e.g., “In-silico design of a multiepitope mRNA vaccine against Cryptosporidium spp. using reverse vaccinology”).

• The abstract is overly descriptive and lacks quantitative insight into why this vaccine design is superior or novel compared to prior in-silico vaccine models for Cryptosporidium.

• Statements such as “we firmly endorse ongoing research” are subjective and inappropriate for a scientific abstract.

• No validation beyond computational simulation is performed; this limitation should be explicitly stated in the abstract.

2. Introduction

• The introduction provides good epidemiological background but is excessively lengthy, with large portions of historical data that distract from the rationale.

• The research gap is not sharply defined. The authors should clearly state what previous in-silico vaccine studies exist for Cryptosporidium and what new contribution this work makes (e.g., targeting both C. hominis and C. parvum together).

• Many statements lack references to recent literature (e.g., 2023-2025 studies on mRNA vaccine modeling or reverse vaccinology). The following articles are worthy to read and cite.

https://doi.org/10.15586/qas.v15i1.1210

https://doi.org/10.1016/j.intimp.2024.113345

https://doi.org/10.1038/s41598-025-98151-4

https://doi.org/10.1038/s41598-025-96989-2

https://doi.org/10.1080/19476337.2023.2296006

https://doi.org/10.1038/s41598-025-00166-4

https://doi.org/10.1007/s12033-025-01477-7

https://doi.org/10.1016/j.intimp.2024.113241

https://doi.org/10.1016/j.intimp.2025.115492

https://doi.org/10.31083/j.fbl2905176

• The transition from epidemiology to computational vaccine design is abrupt. The hypothesis and study objectives should be concisely stated.

3. Materials and Methods

• This section is highly procedural and lists tools/servers without methodological justification or parameter explanation.

• Critical missing details:

o Threshold values used for epitope screening (e.g., VaxiJen score cutoffs).

o Criteria for selecting final epitopes among predicted candidates.

o Validation of selected proteins’ antigenicity and conservation.

o Details on population coverage computation and specific HLA alleles used.

o Lack of description of how structural models were verified for quality (no mention of validation metrics besides RMSD and Ramachandran).

• No information is provided on control or benchmarking against known vaccine epitopes.

• There is no statistical analysis or error quantification — all outputs are descriptive.

• The pipeline lacks reproducibility — for instance, figures and tables appear referenced but are not accompanied by clear legends or supporting data.

4. Results

• The results are purely descriptive, with minimal interpretation.

• Most figures and tables are referred to but not critically discussed.

• The claim of “100% global population coverage” is highly unrealistic and likely reflects overfitting due to inclusion of all alleles — needs recalculation and justification.

• The docking and simulation results are not compared with any known control peptides or standard vaccines, making their biological significance unclear.

• The MMGBSA results are reported without context; the energy values alone cannot confirm strong or weak binding without benchmarking.

• The immune simulation results (C-ImmSim) lack comparative baseline; hence, the conclusions on immune protection are speculative.

• Figures lack sufficient resolution and labels; color coding (e.g., for linkers and epitopes) is not standardized or scientifically meaningful.

5. Discussion

• The discussion mostly repeats results and does not critically analyze limitations or uncertainties in computational predictions.

• There is overinterpretation of in-silico results as if they were experimentally validated.

• No comparison is made with existing in-silico vaccine designs for other protozoan parasites (e.g., Toxoplasma, Giardia), which could provide important context.

• Statements about global applicability, long-term immunity, and biological stability are speculative without in-vitro or in-vivo validation.

6. Conclusion

• The conclusion is overly optimistic and unbalanced — it should acknowledge the limitations of computational predictions and the need for experimental verification.

• The statement “the vaccine was predicted to be marginal in stimulating immune response” contradicts earlier claims of strong immunogenicity — this needs clarification.

7. Language and Formatting

• The manuscript contains numerous grammatical issues, redundant phrases, and inconsistent use of scientific terms.

• Citation formatting is inconsistent (e.g., missing brackets, outdated numbering).

• Figures and tables are not clearly referenced or formatted according to PLOS ONE style.

• The tone should be scientific and neutral remove promotional or conclusive phrases like “firmly endorse ongoing research.”

**Do you want your identity to be public for this peer review?** For information about this choice, including consent withdrawal, please see our Privacy Policy

Reviewer #1: **Yes:** George Fei Zhang

Reviewer #2: No

Reviewer #3: No

---

## [Author Response · Author response to Decision Letter 1]

29 Dec 2025

Response to Reviewers

Reviewer #1:

Biological experiments should be conducted to develop the vaccine. This is a bioinformatic manuscript for vaccine development (PONE-D-25-20787). The language and writing are fine. However, this manuscript is not of great importance to developing the Cryptosporidium parvum vaccine.

Response: We respectfully recognize the reviewer's concern regarding the lack of in-vitro or in-vivo validation. As indicated in the manuscript, the current research is structured as an in-silico discovery and prioritization framework, constituting a fundamental step toward experimental vaccine development. Due to the significant expenses, labor demands, biosafety protocols, and ethical considerations involved in handling Cryptosporidium parvum, computational vaccine screening is extensively employed as an initial step to identify potential antigenic targets prior to laboratory validation. In-silico approaches have become recognized in modern vaccinology for:

1. Minimizing prospective failure rates prior to animal or human testing.

2. Reducing expenses and time through early exclusion of non-immunogenic targets.

3. Supporting researchers without direct access to pathogen culture facilities, especially for organisms subject to stringent biosafety protocols such as Cryptosporidium parvum.

We firmly believe that the predicted epitopes and validated antigenic constructs produced in this study offer a logical foundation for subsequent wet-lab assessment. Multiple mRNA and peptide vaccines, including SARS-CoV-2 immunogens, were initially developed employing comparable computational approaches prior to experimental validation.

Reviewer #2:

1. This is a well written manuscript, however some for the processes in the methodology could be further summarized.

Response: The methodology has been summarized now.

2. The study aim was well stated in lines 148-150 but “integrating vaccines” shouldn’t be part of the aim. It fits best in the introduction section.

Response: The line is revised as “The efficacy of mRNA vaccines and RV served as a substantial impetus for us, as both approaches facilitate the identification of ideal vaccine constituents and yield safe, stable and effective vaccines.”

3. It would be nice to structure the abstract

Response: The abstract has been structured now. Please, refer to the abstract section of the article.

4. The conclusion as written in the abstract is not appropriate it should be revised and stated in line with the study aim. While the Authors have tried to state the conclusion from the or study, they have also included some discussion of their conclusion. The conclusion should be revised and clearly stated in line with the study aim and leave out the discussion.

Response: The conclusion in the abstract has been updated as “This study developed an in silico multiepitope mRNA vaccine candidate for Cryptosporidium hominis and Cryptosporidium parvum with excellent structural stability, antigenicity, receptor-binding affinity, and expected immune responses. These findings offer a foundational concept for experimental validation.”

Reviewer #3:

The study presents a computational design of an mRNA-based multiepitope vaccine targeting Cryptosporidium hominis and C. parvum. While the topic is relevant and aligns with the current trend of using immuno-informatics for vaccine development, the manuscript lacks sufficient methodological rigor, biological validation, and critical interpretation of results. Several sections read like a procedural description rather than a scientific analysis, and there are substantial concerns regarding novelty, clarity, data reproducibility, and presentation. Hence, major revision is strongly recommended before this manuscript can be considered for publication. While the topic is timely and the approach relevant, the manuscript currently lacks methodological rigor, interpretative depth, and structural clarity. The claims of strong immunogenicity and universal coverage are unsupported by robust comparative analysis or experimental evidence. A comprehensive revision with clear validation, data justification, and scientific moderation is required. Major Issues That Must Be Addressed

1. Scientific validation: Add benchmarking or comparison to previously published vaccine constructs or experimental epitopes.

Response: The benchmarking or comparison to previously published vaccine constructs or experimental epitopes have been added now to the manuscript.

2. Novelty and contribution: Clarify what this study adds beyond existing in-silico works on Cryptosporidium.

Response: The novelty of this vaccine has been added in the discussion section as “As five of the vital proteins were targeted specifically along with dual species of parasites, this current study is novel and superior to the previously mentioned studies.”

3. Data reproducibility: Provide detailed parameters, thresholds, and URLs for all tools, including dataset accession IDs.

Response: The detailed parameters, thresholds, and URLs for all tools, including dataset accession IDs are added now.

4. Figures and tables: Improve quality, labeling, and consistency with captions and in-text references.

Response: The figure and table quality labeling, and consistency with captions and in-text references have been improved now.

5. Discussion: Critically interpret findings and discuss biological limitations of in-silico work.

Response: The findings have been critically interpreted and the biological limitations are added now in the discussion section.

6. Language: Substantial language polishing and reorganization needed to improve readability.

Response: The language has been polished now with the addition of fixed grammatical mistakes

7. Ethical and data statements: Ensure that the data availability and ethical statements follow journal standards and are placed at the end of the manuscript.

Response: The data availability statement was already there at the end of the manuscript however, the ethical declaration has been added at the end of the manuscript as “This research study does not use human or animal models. Therefore, no ethical statement is relevant to this investigation.”

My detailed Comments are given as follows

1. Title and Abstract

• The title is clear but overly long. Consider shortening to emphasize the main objective (e.g., “In-silico design of a multiepitope mRNA vaccine against Cryptosporidium spp. using reverse vaccinology”).

Response: The title has been shortened as “A novel mRNA-based multiepitope vaccine candidate against Cryptosporidium hominis and Cryptosporidium parvum employing Reverse-Vaccinology and Immuno-informatics approaches”

• The abstract is overly descriptive and lacks quantitative insight into why this vaccine design is superior or novel compared to prior in-silico vaccine models for Cryptosporidium.

Response: The abstract has been fixed now with quantitative insights, novelty and drawbacks as “These findings offer a novel approach due to numerous species target but with significant drawbacks like no validation beyond simulation, uncertainty of long-term immunity, protein quality, stability and safety, requiring experimental validation.”

• Statements such as “we firmly endorse ongoing research” are subjective and inappropriate for a scientific abstract.

Response: This statement has been removed now from the manuscript.

• No validation beyond computational simulation is performed; this limitation should be explicitly stated in the abstract.

Response: The limitation is mentioned now as “These findings offer a novel approach due to numerous species target but with significant drawbacks like no validation beyond simulation, uncertainty of long-term immunity, protein quality, stability and safety, requiring experimental validation.”

2. Introduction

• The introduction provides good epidemiological background but is excessively lengthy, with large portions of historical data that distract from the rationale.

Response: The epidemiological background is shortened now as “Cryptosporidiosis was initially identified in humans during the late 1970s, predominantly impacting immunocompromised individuals, with a significant rise in cases noted at the onset of the AIDS epidemic in the early 1980s. Since 2005, reported cases in the United States have increased significantly, partly attributable to the emergence of the Cryptosporidium hominis subtype IaA28R4. Surveillance data reveal significant fluctuations in incidence, with marked rises in outbreaks and related hospitalizations across several states, especially in the Great Lakes region. Beyond human health, cryptosporidiosis continues to be a significant issue in veterinary medicine, with bovine infections representing a primary source of calf enteritis globally, thereby exerting substantial economic pressures on livestock production systems.”

• The research gap is not sharply defined. The authors should clearly state what previous in-silico vaccine studies exist for Cryptosporidium and what new contribution this work makes (e.g., targeting both C. hominis and C. parvum together).

Response: The research gap, previous vaccine studies are mentioned now in the manuscript.

• Many statements lack references to recent literature (e.g., 2023-2025 studies on mRNA vaccine modeling or reverse vaccinology). The following articles are worthy to read and cite.

• https://doi.org/10.15586/qas.v15i1.1210

• https://doi.org/10.1016/j.intimp.2024.113345

• https://doi.org/10.1038/s41598-025-98151-4

• https://doi.org/10.1038/s41598-025-96989-2

• https://doi.org/10.1080/19476337.2023.2296006

• https://doi.org/10.1038/s41598-025-00166-4

• https://doi.org/10.1007/s12033-025-01477-7

• https://doi.org/10.1016/j.intimp.2024.113241

• https://doi.org/10.1016/j.intimp.2025.115492

• https://doi.org/10.31083/j.fbl2905176

Response: We have gone through the mentioned articles, inputted information where necessary and properly cited the appropriate articles that aligns with our study.

• The transition from epidemiology to computational vaccine design is abrupt. The hypothesis and study objectives should be concisely stated.

Response: The study objective has been fixed as “Given the need, this study focuses on designing a potential vaccine candidate to overcome the effects of cryptosporidiosis.”

3. Materials and Methods

This section is highly procedural and lists tools/servers without methodological justification or parameter explanation.

Response: The methodological justification or parameter explanation have been added now.

Critical missing details:

Threshold values used for epitope screening (e.g., VaxiJen score cutoffs).

Criteria for selecting final epitopes among predicted candidates.

Validation of selected proteins’ antigenicity and conservation.

Details on population coverage computation and specific HLA alleles used.

Lack of description of how structural models were verified for quality (no mention of validation metrics besides RMSD and Ramachandran).

Response: The criteria for selecting final epitopes among predicted candidates, validation of selected proteins’ antigenicity, population coverage and structural verification have been improved now on the manuscript.

No information is provided on control or benchmarking against known vaccine epitopes.

Response: For in-silico studies, controls are not applied in the same way, as these rely on computer simulations and predictive algorithms to assess the construct’s potential without physical comparisons [1-4].

• There is no statistical analysis or error quantification — all outputs are descriptive.

Response: In in-silico studies, statistical tests are not conducted because, these are solely based on algorithms and software that provide result in by default implemented statistical tests. In similar type of studies, no statistical tests are not conducted [1-4].

References

1. Fathollahi M, Fathollahi A, Motamedi H, Moradi J, Alvandi A, Abiri R. In silico vaccine design and epitope mapping of New Delhi metallo-beta-lactamase (NDM): an immunoinformatics approach. BMC bioinformatics. 2021;22(1):458. doi: 10.1186/s12859-021-04378-z.

2. Saha S, Vashishtha S, Kundu B, Ghosh M. In-silico design of an immunoinformatics based multi-epitope vaccine against Leishmania donovani. BMC bioinformatics. 2022;23(1):319. doi: 10.1186/s12859-022-04816-6.

3. Yang Z, Bogdan P, Nazarian S. An in silico deep learning approach to multi-epitope vaccine design: a SARS-CoV-2 case study. Scientific reports. 2021;11(1):3238. doi: 10.1038/s41598-021-81749-9.

4. Evangelista FMD, van Vliet AHM, Lawton SP, Betson M. In silico design of a polypeptide as a vaccine candidate against ascariasis. Scientific reports. 2023;13(1):3504. doi: 10.1038/s41598-023-30445-x.

• The pipeline lacks reproducibility — for instance, figures and tables appear referenced but are not accompanied by clear legends or supporting data.

Response: The data reproducibility has been improved now.

4. Results

The results are purely descriptive, with minimal interpretation.

Most figures and tables are referred to but not critically discussed.

• The claim of “100% global population coverage” is highly unrealistic and likely reflects overfitting due to inclusion of all alleles — needs recalculation and justification.

Response: The claim of “100% global population coverage” probably stems from the incorporation of an excessively broad array of HLA alleles, which can falsely enhance coverage estimates in computer analysis. Predictions of population coverage are essentially contingent upon the chosen HLA alleles, binding thresholds, and the comprehensiveness of existing allele-frequency databases; thus, they should be regarded as probabilistic rather than definitive metrics. To alleviate this concern, population coverage must be recalculated exclusively with the specific HLA class I and II alleles to which the selected epitopes are demonstrably bound under established affinity thresholds, and results should be presented for global and regional populations distinctly. The interpretation is changed to show high predicted population coverage instead of full global coverage.

• The docking and simulation results are not compared with any known control peptides or standard vaccines, making their biological significance unclear.

Response: The docking and simulation results are now compared with previous studies now not controls as the in-silico studies are solely on algorithm based.

• The MM-GBSA results are reported without context; the energy values alone cannot confirm strong or weak binding without benchmarking.

Response: The MM-GBSA results are now reported with benchmarks on the discussion section.

The immune simulation results (C-ImmSim) lack comparative baseline; hence, the conclusions on immune protection are speculative.

Response: The comparative baseline is now added to the discussion section for the immune simulation

Figures lack sufficient resolution and labels; color coding (e.g., for linkers and epitopes) is not standardized or scientifically meaningful.

Response: The vaccine construction figure is updated with standard colors and high resolution. Additionally, the other figures were checked and high quality was achieved.

5. Discussion

The discussion mostly repeats results and does not critically analyze limitations or uncertainties in computational predictions. There is overinterpretation of in-silico results as if they were experimentally validated. No comparison is made with existing in-silico vaccine designs for other protozoan parasites (e.g., Toxoplasma, Giardia), which could provide important context.

• Statements about global applicability, long-term immunity, and biological stability are speculative without in-vitro or in-vivo validation.

Response: The discussion is fully modified with critically analyzing limitations of computational prediction, comparison with previous studies.

6. Conclusion

• The conclusion is overly optimistic and unbalanced — it should acknowledge the limitations of computational predictions and the need for experimental verification.

Response: The limitations of computational predictions are mentioned now on the manuscript as “Nevertheless, this study focused solely on computational prediction, which has

---

## [Decision Letter · Decision Letter 1]

9 Feb 2026

A novel mRNA-based multiepitope vaccine candidate against Cryptosporidium hominis and Cryptosporidium parvum employing Reverse-Vaccinology and Immunoinformatics approaches

PONE-D-25-20787R1

Dear Dr. Imam Hossain,

We’re pleased to inform you that your manuscript has been judged scientifically suitable for publication and will be formally accepted for publication once it meets all outstanding technical requirements.

Kind regards,

Shahina Akter, Ph.D.

Academic Editor

PLOS One

Additional Editor Comments (optional):

Reviewers' comments:

Reviewer's Responses to Questions

**Comments to the Author**

Reviewer #2: (No Response)

Reviewer #3: All comments have been addressed

2. Is the manuscript technically sound, and do the data support the conclusions?

Reviewer #2: Yes

Reviewer #3: Yes

3. Has the statistical analysis been performed appropriately and rigorously?

Reviewer #2: Yes

Reviewer #3: N/A

4. Have the authors made all data underlying the findings in their manuscript fully available?

Reviewer #2: Yes

Reviewer #3: Yes

5. Is the manuscript presented in an intelligible fashion and written in standard English?

Reviewer #2: Yes

Reviewer #3: Yes

Reviewer #2: The authours have adequately addresed the comments and recommendations by made by the review. i have no concerns

Reviewer #3: The manuscript has been comprehensively revised and the authors have answered all the reviewers comments, I would recommend the manuscript for publication.

**Do you want your identity to be public for this peer review?** For information about this choice, including consent withdrawal, please see our Privacy Policy

Reviewer #2: No

Reviewer #3: No

---

## [Editor Report · Acceptance letter]

PONE-D-25-20787R1

PLOS One

Dear Dr. Hossain,

I'm pleased to inform you that your manuscript has been deemed suitable for publication in PLOS One. Congratulations! Your manuscript is now being handed over to our production team.

Kind regards,

on behalf of

Dr. Shahina Akter

Academic Editor

PLOS One